# The NAD salvage pathway in mesenchymal cells is indispensable for skeletal development in mice

Aaron Warren [1,8], Ryan M. Porter [1,2,3,8], Olivia Reyes-Castro[1], Md Mohsin Ali[1], Adriana Marques-Carvalho[4], Ha-Neui Kim [1,2], Landon B. Gatrell[1], Ernestina Schipani[5], Intawat Nookaew [2,6], Charles A. O'Brien[1,2,3], Roy Morello [2,3,7] & Maria Almeida [1,2,3] ✉

NAD is an essential co-factor for cellular energy metabolism and multiple other processes. Systemic NAD$^+$ deficiency has been implicated in skeletal deformities during development in both humans and mice. NAD levels are maintained by multiple synthetic pathways but which ones are important in bone forming cells is unknown. Here, we generate mice with deletion of Nicotinamide Phosphoribosyltransferase (*Nampt*), a critical enzyme in the NAD salvage pathway, in all mesenchymal lineage cells of the limbs. At birth, *Nampt*$^{\Delta Prx1}$ exhibit dramatic limb shortening due to death of growth plate chondrocytes. Administration of the NAD precursor nicotinamide riboside during pregnancy prevents the majority of in utero defects. Depletion of NAD post-birth also promotes chondrocyte death, preventing further endochondral ossification and joint development. In contrast, osteoblast formation still occurs in knockout mice, in line with distinctly different microenvironments and reliance on redox reactions between chondrocytes and osteoblasts. These findings define a critical role for cell-autonomous NAD homeostasis during endochondral bone formation.

Nicotinamide adenine dinucleotide (NAD) is a critical cofactor for cellular energy metabolism and numerous cellular processes, such as cell division, DNA damage repair, and mitochondrial function[1,2]. In its oxidized state, NAD$^+$ and its phosphorylated version NADP$^+$ are able to accept two electrons, making this molecule essential for redox homeostasis. In this role, NAD$^+$ is central for the catabolism of carbon sources in glycolysis, TCA cycle, β-oxidation, and glutaminolysis. Conversely, the redox pair NADP$^+$/NADPH is primarily used for anabolic processes such as lipid and cholesterol synthesis. NAD$^+$ also acts as a degradation substrate for numerous enzymes, such as

sirtuins, PARPs, and cyclic ADP-ribose synthetases such as CD38 and CD157[3,4].

NAD is synthesized from dietary precursors such as tryptophan and vitamin B3 (niacin) via several pathways[5,6]. Vitamin B3 is a collective term for nicotinic acid (NA), nicotinamide (NAM), and nicotinamide riboside (NR)[3]. The kynurenine pathway, also known as the de novo pathway, produces NAD from tryptophan. The Preiss–Handler and salvage pathways produce NAD from NA and NAM, respectively[7]. The existence of different pathways leading to NAD$^+$ production raises questions regarding the relative importance of each pathway to

[1]Division of Endocrinology and Metabolism, University of Arkansas for Medical Sciences, Little Rock, AR, USA. [2]Center for Musculoskeletal Disease Research, University of Arkansas for Medical Sciences, Little Rock, AR, USA. [3]Department of Orthopedic Surgery, University of Arkansas for Medical Sciences, Little Rock, AR, USA. [4]CNC—Center for Neuroscience and Cell Biology, University of Coimbra, UC-Biotech, Biocant Park, Cantanhede, Portugal. [5]Department of Orthopaedic Surgery, University of Pennsylvania, Philadelphia, PA, USA. [6]Department of Biomedical Informatics, University of Arkansas for Medical Sciences, Little Rock, AR, USA. [7]Department of Physiology and Cell Biology, University of Arkansas for Medical Sciences, Little Rock, AR, USA. [8]These authors contributed equally: Aaron Warren, Ryan M. Porter. ✉e-mail: schullermaria@uams.edu

various tissues. For instance, the kynurenine pathway of NAD synthesis seems to occur predominantly in the liver, as several enzymes of this pathway are not expressed in most other tissues[8]. Furthermore, plasma levels of most NAD+ precursors are low and, most likely, unable to systematically sustain high NAD+ production rates. The intracellular levels of NAD+ are determined by the cellular redox state (i.e., NAD+/NADH) and by the balance between NAD+ biosynthesis and degradation. The NAD-dependent sirtuins, PARPs, and cyclic ADP-ribose synthetases metabolize NAD+ into NAM. The NAM generated in these processes can be recycled to NAD via the salvage pathway[9].

Multiple studies using genetic and pharmacological tools have elucidated that a decrease in NAD has a causative role in several diseases of aging, including osteoporosis[9–11]. In addition, skeletal deformities occurring during development have been associated with NAD deficiency[12]. The majority of the mammalian skeleton, including long bones in the limbs, forms through a process known as endochondral ossification[13]. During embryonic development, mesenchymal cells descend from the lateral plate mesoderm and form condensations within the limb bud. These cells differentiate into chondrocytes which form and expand an avascular structure known as the cartilage anlage[14]. Initially, chondrocytes proliferate at a high rate and produce an extracellular matrix rich in type II collagen and proteoglycans. Next, they exit the cell cycle and differentiate into hypertrophic cells that produce type X collagen. Vascular invasion into the hypertrophic zone allows for the formation of a bone marrow cavity called the primary ossification center and replacement of cartilage by bone[15]. The cartilage template continues to extend from both bone ends (epiphyses) due to proliferation and hypertrophy of chondrocytes comprising the growth plates. Indeed, these structures contain three distinct layers of resting, proliferating, and hypertrophic chondrocytes. After birth, a zone of hypertrophic chondrocytes then develops at the center of the epiphysis, followed by blood vessel invasion and deposition of bone matrix, initiating the formation of the secondary ossification center[16]. Alternatively, some bones, such as the flat bones of the skull, develop through mesenchymal intramembranous ossification, a process in which mesenchymal cells differentiate directly into osteoblasts.

Disruption of the normal sequence of cartilage development causes a multitude of skeletal dysplasias in humans[17]. Bi-allelic loss-of-function variants in enzymes of the kynurenine pathway cause congenital malformations in several organs, including the skeleton, as a consequence of NAD deficiency[18]. These malformation phenotypes have been classified as Congenital NAD Deficiency Disorder also called vertebral, cardiac, renal, and limb defects syndrome (Online Mendelian Inheritance in Man [OMIM] numbers 617660 and 617661, respectively). Likewise, restricted maternal intake of NAD precursors during pregnancy in mice leads to malformations in organs such as the heart, kidney, and skeleton[19]. Nonetheless, the systemic nature of these effects precludes any conclusions about direct effects of NAD in bone cells. To elucidate the contribution of NAD in specific bone cell populations to skeletal development, we generated mice with conditional deletion of *Nampt* – an essential enzyme in the NAD salvage pathway – in cells of the mesenchymal lineage. We found that the NAD salvage pathway in osteochondroprogenitors is indispensable for embryonic skeletal development as well as for postnatal skeletal growth and homeostasis.

## Results

### Loss of *Nampt* in mesenchymal progenitors impairs skeletal development

To examine the role of the NAD salvage pathway in skeletal development, we used Prx1-Cre transgenic mice to delete *Nampt* in all mesenchymal cells of the limb, including precursors of chondrocytes and osteoblasts (*Nampt*^ΔPrx1^)[20]. Prx1-Cre also targets precursors for the calvarium and the sterna but does not impact the axial skeleton. Cre recombinase activity starts early in development and is greater in forelimbs than hindlimbs[20]. *Nampt*^ΔPrx1^ mice were born at the expected

Mendelian ratio but displayed severe shortening of the limbs at birth (Fig. 1a). All mice died a few days after birth for yet unknown reasons. Whole-mount skeletal staining at P3 confirmed that all limb skeletal elements were markedly shorter in *Nampt*^ΔPrx1^ embryos compared to controls (Fig. 1b). The forelimb was more severely impacted, showing a very rudimentary stylopod (humerus) and zeugopod (radius and ulna) and missing the autopod (paw), most likely due to the higher penetrance of Prx1-Cre at this site (Fig. 1b). Likewise, in the hindlimb, autopod formation was absent (Fig. 1c). Immunostaining confirmed the loss of *Nampt* expression within the growth plate chondrocytes of P2 *Nampt*^ΔPrx1^ hindlimbs, in contrast to surrounding muscle cells that are not targeted by Prx1-Cre (Fig. 1d). Similar to the limbs, *Nampt* deletion caused shortening of the sternum (Fig. 1e). In contrast, no obvious defects were seen in the craniofacial skeleton (Fig. 1f).

### Loss of *Nampt* compromises cartilage homeostasis

We next performed histology of limbs and sternum to evaluate the cellular defects in the bones of *Nampt*^ΔPrx1^ mice. H&E staining of femoral sections of littermate control mice at P2 revealed the typical zones present in epiphyseal cartilage at this age, namely the resting, proliferative and hypertrophic zones (Fig. 2a). The growth plate of Nampt^ΔPrx1^ mice exhibited low chondrocyte number with an almost total absence of the resting zone. Islets of proliferating chondrocytes were present, as well as hypertrophic chondrocytes. Deletion of *Nampt* also caused major deficits in extracellular matrix, as shown by a faint, mainly pericellular Safranin O staining for proteoglycans that did not closely overlay with deposition of type II collagen. Very similar morphological findings were seen in the sternum (Supplementary Fig. 1). To examine whether the decrease in cellularity was due to cell death, we performed TUNEL staining. Rare TUNEL-positive cells were seen in the growth plate of control mice at this age (Fig. 2b). In contrast, *Nampt*^ΔPrx1^ mice exhibited numerous positive cells. Histological analysis also revealed that synovial joint formation was disrupted in *Nampt*^ΔPrx1^ mice. Specifically, the joints between the femur and tibia and tibia and fibula were fused (Fig. 2c).

### Administration of NR to pregnant mothers greatly attenuated the skeletal developmental defects caused by *Nampt* deletion

To confirm that the defects seen in *Nampt*^ΔPrx1^ mice were due to a decrease in NAD, we administered the NAD precursor NR to mice during pregnancy. NR is transformed into nicotinamide mononucleotide (NMN), which is directly converted to NAD by the salvage pathway, bypassing the need for Nampt (Fig. 3a). A dose of about 400 mg/kg NR was not sufficient to alter the developmental defects of *Nampt*^ΔPrx1^ mice (Supplementary Fig. 2). However, administration of 1000 mg/kg NR to the dams greatly attenuated the skeletal defects in *Nampt*^ΔPrx1^ mice, as determined at P2. Specifically, the forelimbs were well formed and contained all elements including stylopod, zeugopod, and autopod (Fig. 3b). The lower limbs contained all digits (Fig. 3c). Immunohistochemistry confirmed the deletion of Nampt in chondrocytes of the hindlimbs (Fig. 3d) of *Nampt*^ΔPrx1^ mice. As expected, Nampt expression remained unaffected in the adjacent muscle tissue. Safranin O staining for proteoglycans was highly similar between control and *Nampt*^ΔPrx1^ rescued mice (Fig. 3e). H&E staining of the humerus revealed that the growth plate of *Nampt*^ΔPrx1^ mice that received NR during gestation was practically indistinguishable from control mice (Supplementary Fig. 3). In addition, NR administration prevented joint malformations, as shown both in the hindlimbs and forelimbs (Fig. 3e and Supplementary Fig. 3). Nevertheless, both limbs (Fig. 3b, c) and sternum (Supplementary Fig. 4) of *Nampt*^ΔPrx1^ rescued mice remained shorter than littermate controls and the menisci were not well developed (Fig. 3e). In addition, *Nampt*^ΔPrx1^ mice exhibited some TUNEL-positive cells, particularly at the growth plate surfaces of the knees (Fig. 3f) and other synovial joints (Supplementary Fig. 5). In contrast, TUNEL positivity was only observed in the perichondrium/synovium of control

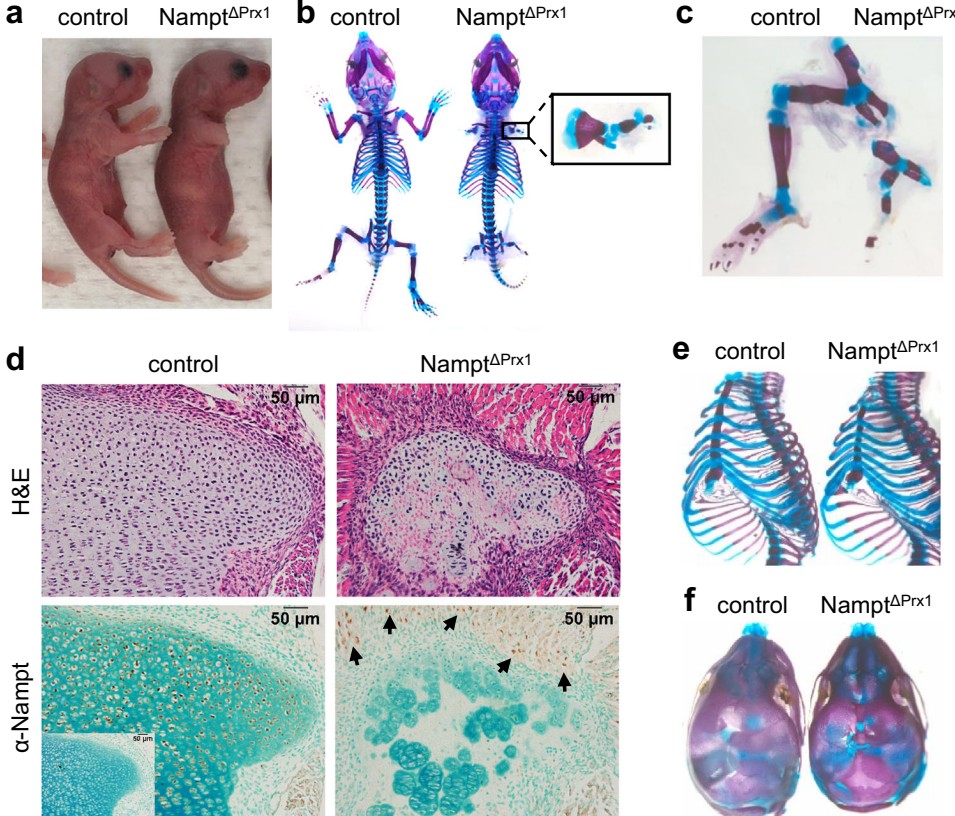

**Fig. 1 | Nampt deletion in mesenchymal lineage cells severely compromises skeletal development. a** P2 Nampt[f/+] (control) and Nampt[f/f];Prx1-Cre (Nampt[ΔPrx1]) mice. **b** Whole-mount alcian blue-alizarin red S staining of newborn skeletons, and (**c**) hindlimbs. **d** Serial paraffin sections from hindlimb growth plates showing H&E or Nampt immunostaining (inset = isotype control). Arrows indicate immunopositive cells within adjacent muscle ($n = 3$ mice/group). **e** Alcian blue-alizarin red S staining of ribcage, and (**f**) head. Four independent litters of mice were obtained and showed similar effects.

(Nampt[f/+]) limbs, as previously described[21]. Importantly, NR administered to dams is not transmitted to milk[22] therefore, after birth the pups do not receive NR supplementation when suckling. Thus, the appearance of dead cells in Nampt[ΔPrx1] rescued mice at P2 is an early consequence of post-natal NAD depletion.

### The NAD salvage pathway is indispensable for epiphyseal bone formation during post-natal development

The ability of NR to rescue most of the pre-natal developmental defects allowed mice to ambulate and survive post-birth. This provided the opportunity to examine the dependency of the growth plate on Nampt during the phase in which epiphyseal bone development begins. To this end, pups from which NR was removed at birth were evaluated at P7 and P28. At P7, a secondary hypertrophic zone was present within the epiphysis in control mice, prefiguring the secondary ossification center (Fig. 4a). No such hypertrophy was present in the Nampt[ΔPrx1] mice. Instead, numerous empty chondrocyte lacunae were seen throughout the epiphysis. TUNEL staining revealed extensive chondrocyte death (Fig. 4b). In addition, the articular cartilage and menisci appeared hypocellular in Nampt[ΔPrx1] mice (Fig. 4c). Furthermore, mRNA levels of chondrocyte-specific genes, such as *Acan*, *Col2a1*, and *Col10a1*, as well as *Mmp13* and *Runx2*, were greatly decreased in cartilage from Nampt[ΔPrx1] mice (Fig. 4d). In contrast, expression of *Vegfa* was not affected.

At P28, control mice exhibited a completely developed and mineralized secondary ossification center, as shown using μCT scans (Fig. 5a). Despite administration of NR during gestation, deletion of *Nampt* completely prevented the post-natal development of the epiphysis and bone growth in the femur and all other bones of the limbs. Histology of the tibia and proximal and distal tibia joints revealed that the epiphysis remained cartilaginous in 28 day-old Nampt[ΔPrx1] mice (Fig. 5b). Yet, a clear bone collar with marrow elements formed within the tibial diaphysis from the primary ossification center during gestation. Because Prx1-Cre targets all cells of the osteoblast lineage, we also examined osteoblasts and bone formation at the endocortical surface of long bones in 28-day-old mice. At this age of rapid growth, osteoblast covered practically all endocortical surfaces in control mice (Fig. 5c). As expected, double labels were abundant on these surfaces (Fig. 5d). Surprisingly, endocortical surfaces of Nampt[ΔPrx1] mice were also covered with active osteoblasts and double labels in a manner indistinguishable from the controls. Quantification of osteoblast number and bone formation rate revealed no differences between the two genotypes (Fig. 5c, d). However, the major differences in size and shape of the bones make it difficult to accurately determine possible consequences of *Nampt* deletion in osteoblastic cells.

To determine whether osteoblasts depend on Nampt during bone development, we generated Nampt[ΔOsx1] mice. Osx1-Cre targets all osteoblast precursors, osteoblasts, and osteocytes as well as a small number of proliferating and hypertrophic chondrocytes[23]. Nampt[ΔOsx1] mice were grossly undistinguishable from Osx1-Cre control littermates at birth and up to 4 weeks-of-age (Fig. 5e), in stark contrast with the findings obtained with Nampt[ΔPrx1] mice (Fig. 1). Evaluations of body weight (Fig. 5f), as well as total and femur dual X-ray absorptiometry (DXA) bone mineral density (BMD) at 4 weeks (Fig. 5g) revealed no differences between mice of the two genotypes.

### Chondrocytes exhibit high expression of metabolic genes that require NADP+/NADPH and NAD+/NADH redox reactions

A potential explanation for the different reliance on the NAD salvage pathway between chondrocytes and osteoblasts is that chondrocytes

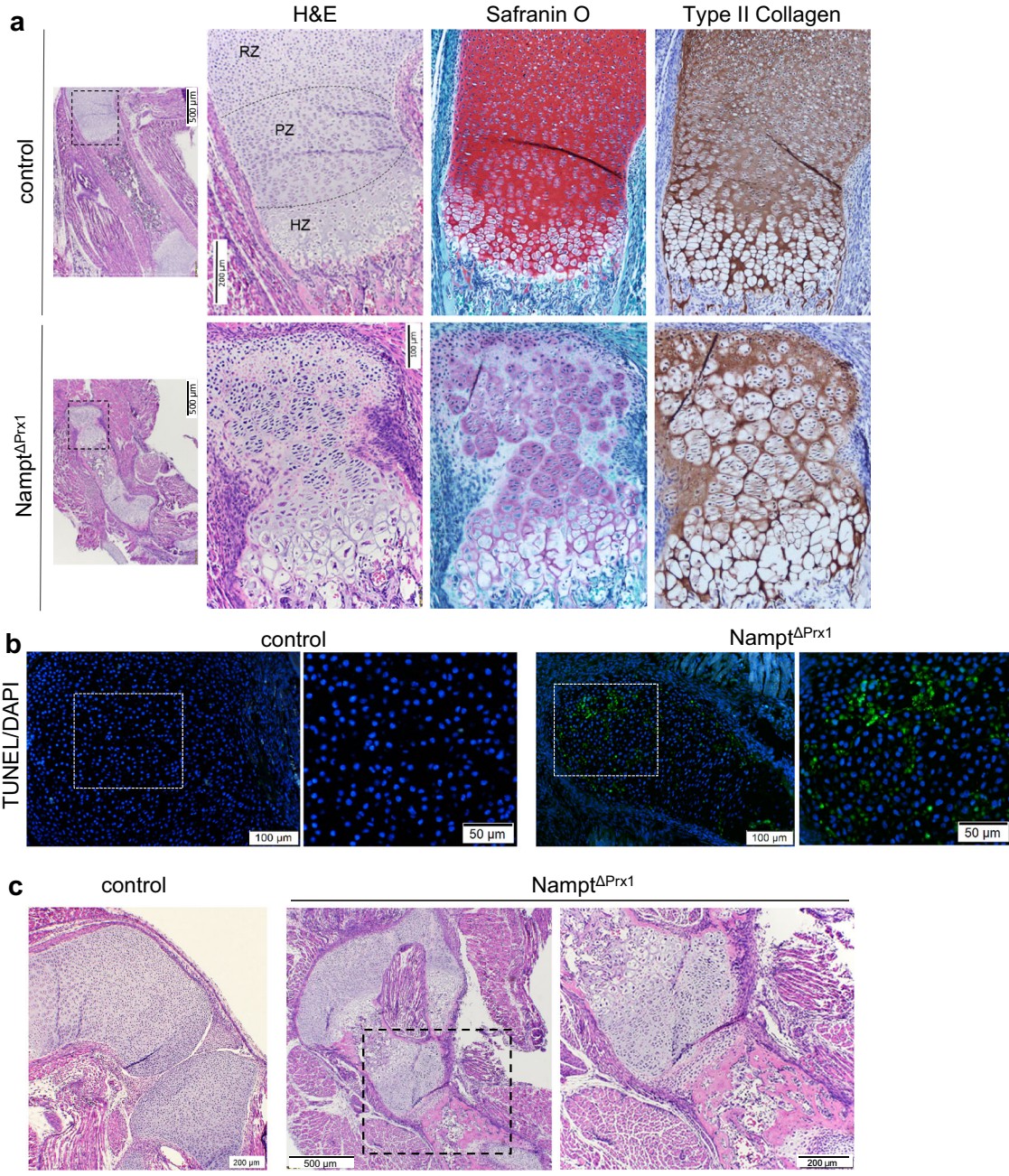

**Fig. 2 | Nampt deletion promotes growth plate chondrocyte apoptosis and disrupts joint formation. a** H&E, Safranin O/Fast Green, or type II collagen immunostaining of histological sections of P2 Nampt$^{f/+}$ (control) and Nampt$^{f/f}$;Prx1-Cre (Nampt$^{\Delta Prx1}$) femurs (n = 3 mice/group). The dotted black boxes are enlarged and shown to the right. RZ resting zone, PZ proliferative zone, HZ hypertrophic zone (n = 3 mice/group). **b** Cell death by TUNEL assay in growth plate of femurs. TUNEL positive signal appears in green. **c** Histological sections as in panel a illustrating the fused joints in the hindlimb of Nampt$^{\Delta Prx1}$ mice. Four independent litters of mice were obtained and showed similar effects.

reside in an avascular microenvironment, making them hypoxic and dependent on nutrient diffusion. To explore the idea that reduced access to circulating nutrients compared to other cell types might underlie the reliance on the NAD salvage pathway, we examined metabolic gene expression patterns using publicly available single cell RNA sequencing datasets from 2 publications in which the authors used mesenchymal cell-enriched samples from femurs of young adult mice[24,25] (Fig. 6a). Combined analysis of the non-hematopoietic cells in this datasets uncovered 10 cell types including growth plate chondrocytes, articular chondrocytes, and osteoblasts characterized by high expression of specific genes such as *Col2a1*, *Prg4*, and *Bglap*, respectively (Supplementary Fig. 6). Based on the differences in

transcription of genes related to metabolism between chondrocytes and the other cell types, we identified metabolic hot spots using reporter metabolite analysis[26]. This analysis revealed that growth plate chondrocyte metabolism was particularly enriched in processes that involve redox reactions dependent on NAD$^+$/NADH and NADP$^+$/NADPH when compared to osteoblasts or the other mesenchymal cell populations present in bone (Fig. 6b). One of the metabolic processes that is highly dependent on both NAD$^+$/NADH and NADP$^+$/NADPH redox reactions is the synthesis of glycosaminoglycans (GAGs) – the most abundant matrix components of chondrocytes[27]. As anticipated, chondrocytes were enriched in metabolic hotspots for chondroitin sulfate, a major component of their matrix. NADPH is also required for

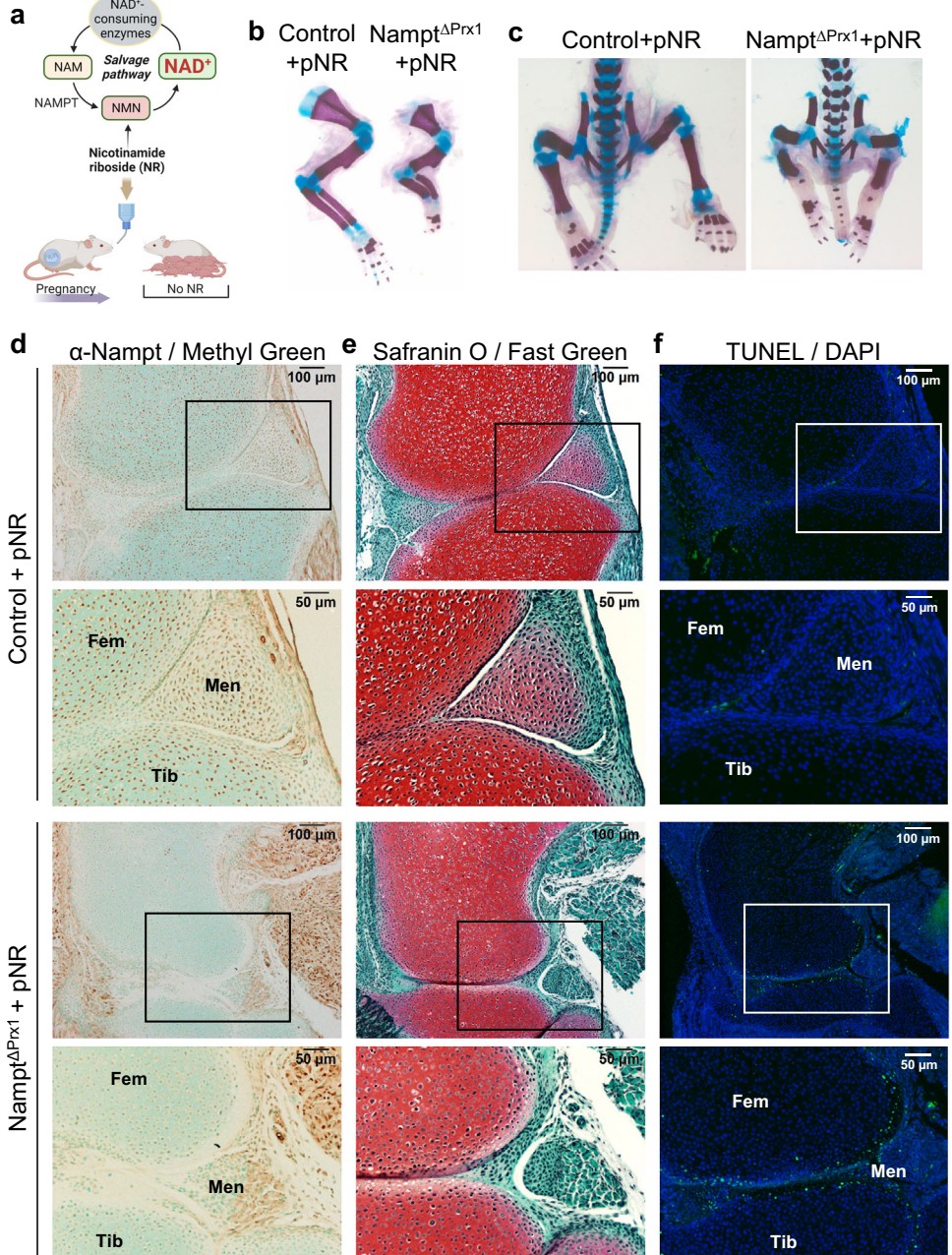

**Fig. 3 | NR administration during pregnancy rescued most of the skeletal defects caused by Nampt deletion. a** NR was administered in the drinking water to dams during pregnancy. NR is converted to NMN and NAD, bypassing the need for Nampt in the salvage pathway. Figure created using BioRender. **b** Whole-mount alcian blue-alizarin red S staining of forelimbs and (**c**) hindlimbs ($n = 3$); pre-natal NR (pNR). **d**, **e** Serial sections from P2 Nampt$^{f/+}$ (control) and Nampt$^{f/f}$;Prx1-Cre (Nampt$^{\Delta Prx1}$) knees (representative of 2–3 pups per genotype) showing immunostaining for Nampt (**d**), Safranin O staining for proteoglycans (**e**), or TUNEL staining (green fluorescence) for dead cells (**f**). Fem femur, Tib tibia, Men Meniscus.

anabolic processes such as lipids and cholesterol synthesis. Lipogenesis was upregulated in growth plate chondrocytes as indicated by the numerous hotspots for acyl-CoenzymeA (CoA) derivatives of long chain polyunsaturated fatty acids (PUFAs). This chemical modification is required for otherwise non-reactive fatty acids to participate in biosynthetic or catabolic pathways. Indeed, through a series of desaturation and elongation reactions PUFA-CoA can be converted into biologically active products[28,29]. Multiple genes responsible for lipid synthesis and processing were upregulated in chondrocytes, including fatty acid synthetase (*Fasn*) (Fig. 6c), which uses NADPH to convert acetyl-CoA into palmitate and is the only enzyme in mammals capable of converting metabolized sugar into a fatty acid. The delta-5 and

delta-6 desaturases (D5D and D6D), encoded by *Fads1* and *Fads2*, respectively, were also upregulated in chondrocytes. These enzymes are responsible for highly unsaturated fatty acid synthesis and regenerate NAD$^+$ from NADH[30]. Furthermore, D5D/D6D provide a mechanism for glycolytic NAD$^+$ recycling that permits ongoing glycolysis and cell viability when the cytosolic NAD$^+$/NADH ratio is reduced, analogous to lactate production[31]. Cholesterol synthesis in the endoplasmic reticulum utilizes both NADH and NADPH as the electron source at multiple steps[32]. Multiple enzymes responsible for cholesterol uptake and synthesis were also upregulated in chondrocytes (Fig. 6c).

As previously shown[33], chondrocytes exhibited up-regulation of genes involved in glycolysis (Fig. 6c), the major energy pathway in

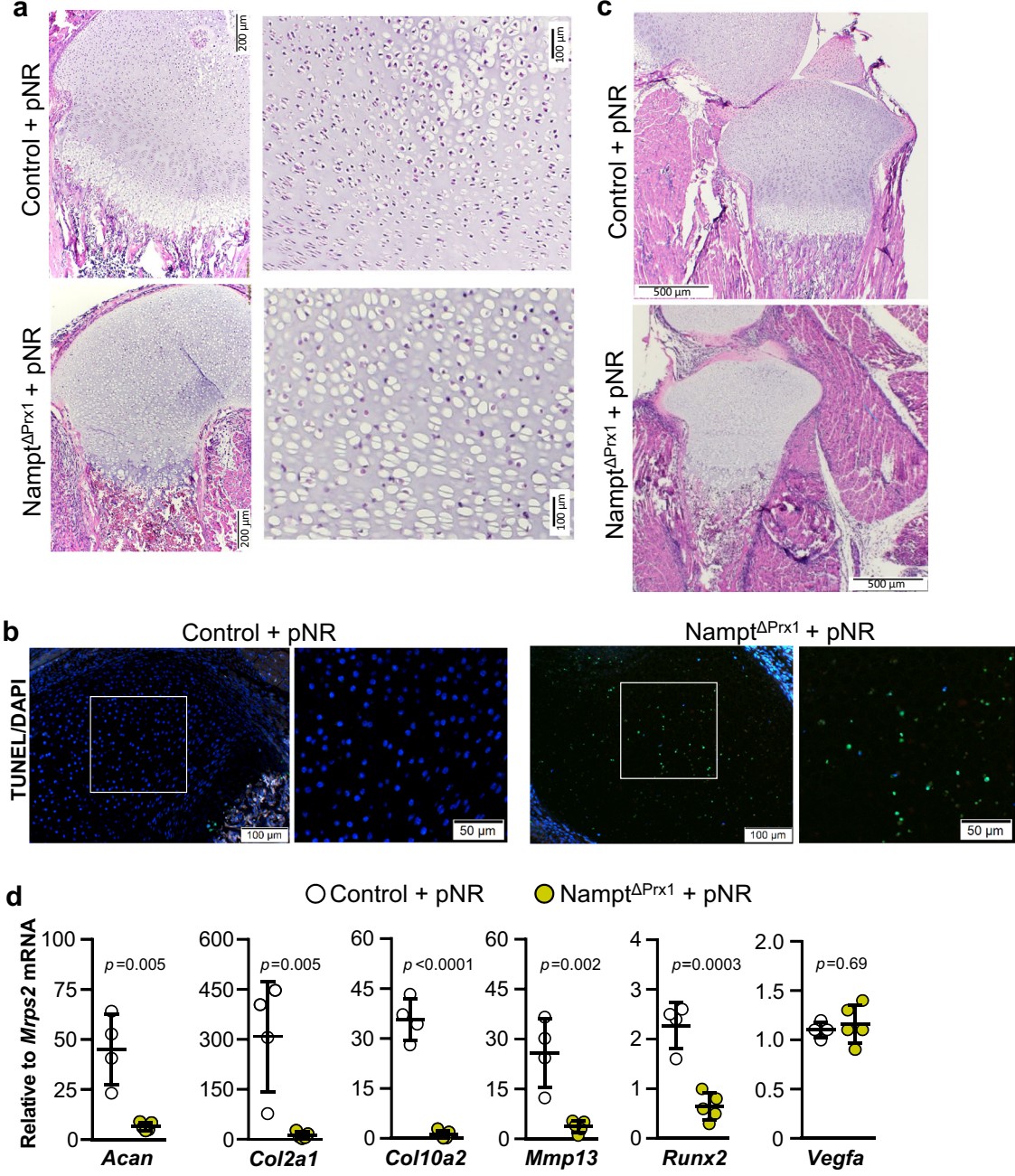

**Fig. 4 | The NAD salvage pathway in mesenchymal cells is indispensable for formation of secondary ossification center and joint development post-birth.** NR was administered in the drinking water to dams during pregnancy (pNR) and progeny was examined at P7. **a** H&E staining performed on histological sections of Nampt[f/+] (control) and Nampt[f/f];Prx1-Cre (Nampt[ΔPrx1]) femurs (*n* = 3). The black boxes are enlarged and shown on the right. **b** Cell death by TUNEL assay in growth plate of femurs. **c** Histological sections as in panel a illustrating the damaged superficial cartilage at the knee joint in Nampt[ΔPrx1] mice. **d** Expression of the indicated genes in cartilage dissected from the hind limbs by quantitative RT-PCR (*n* = 4 control and 5 mice/group). Lines represent mean ± SD, *p* values by two-sided Student t-test. Source data are provided as a Source Data file.

these cells. During glycolysis, high energy hydride ions are captured from glucose by enzymes that utilize NAD$^+$ as an electron acceptor forming NADH. In hypoxia, conversion of pyruvate to lactate by lactate dehydrogenase (Ldha) also converts NADH to NAD$^+$, thereby restoring NAD$^+$ levels in the cytosol. Overall, the metabolic differences between chondrocytes and other bone cells, highlighted in this analysis, are in line with previously described metabolic adaptations to hypoxia[34]. Indeed, besides the changes in metabolism and ATP production, cells also adjust to hypoxia by altering the biosynthesis of macromolecules such as lipids.

## Hypoxia does not render chondrocytes more sensitive to Nampt inhibition

The metabolic adaptation of chondrocytes to hypoxia, namely upregulation of glycolysis, lipid, and cholesterol synthesis[30] might render them more susceptible to the loss of Nampt. To examine this possibility, we used the pharmacological Nampt inhibitor FK866 (1 µM) in primary chondrocytes cultured under normoxia (20% $O_2$) or hypoxia (1% $O_2$). Chondrocytes were left untreated for 2 d to adjust to hypoxia and treated with FK866 for an additional 1 or 2 d. After a total of 3 and 4 d in hypoxia, vehicle-treated cells had lower levels of NAD than cells in

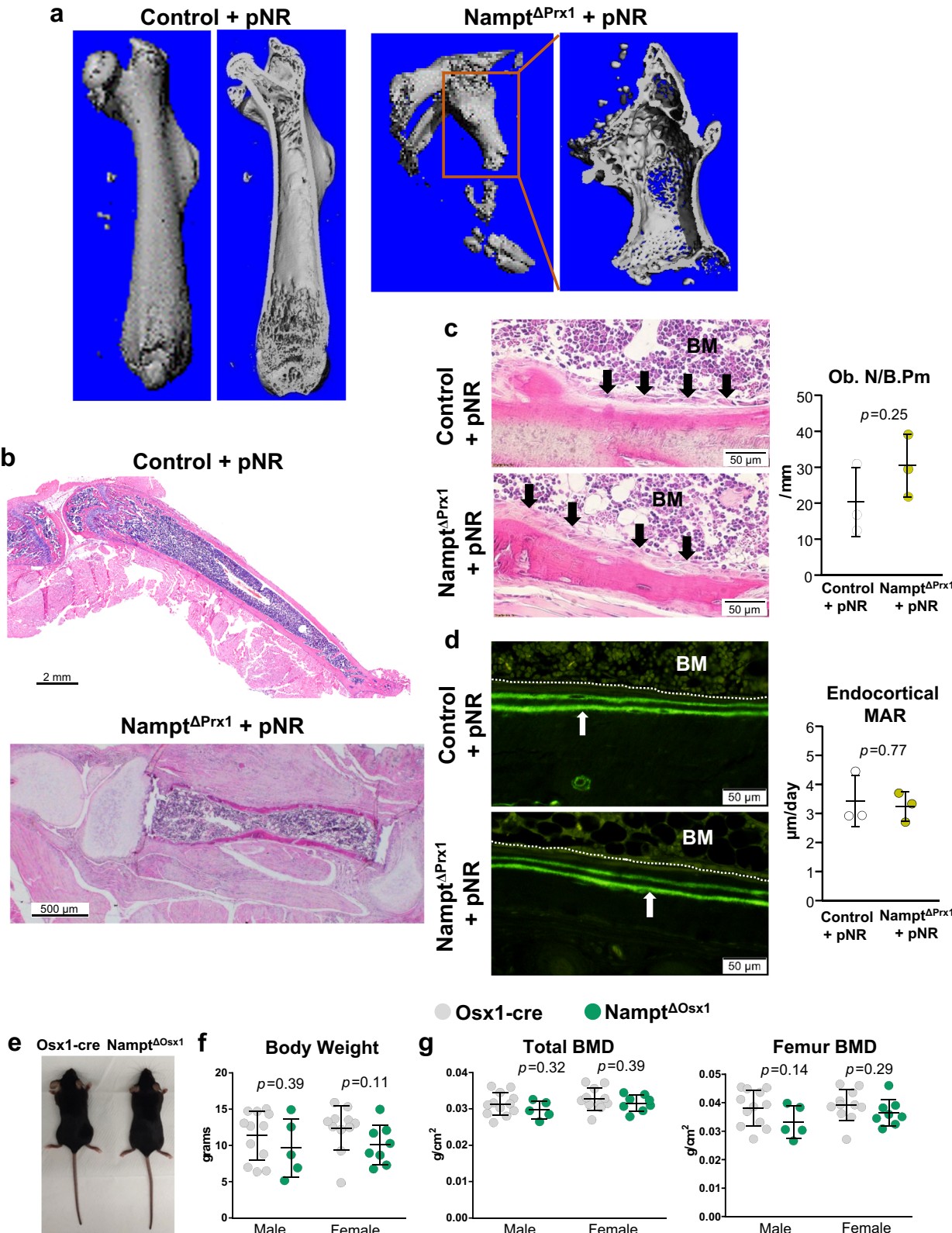

normoxia (Fig. 7a, b), most likely reflecting a decrease in mitochondria respiration. As previously shown[35], cells in hypoxia had higher levels of NADH/NAD ratio (Fig. 7c). The levels of NAD+ were practically undetectable in FK866-treated cells in either normoxia or hypoxia conditions (Fig. 7a, b). We also examined the effect of NAD+ depletion on ATP production after treatment with FK866. Vehicle-treated chondrocytes cultured in hypoxia for 3 d had decreased ATP levels (Fig. 7d). However,

after 4 d in hypoxia the levels of ATP in hypoxic cells were similar to the ones seen in cells cultured in normoxia (Fig. 7e). These changes in ATP most likely reflect the metabolic adaptation of cells to hypoxia, i.e., decreased mitochondrial- and increased glycolysis-dependent energy production. Nampt inhibition also caused ATP depletion, but this effect was seen only after 2 d of FK866 treatment and was similar between cells in normoxia and in hypoxia (Fig. 7d, e). Measurements of lactate

**Fig. 5 | The NAD salvage pathway in mesenchymal cells is required for post-natal endochondral, but not intramembranous, bone formation. a** MicroCT images from femur of Nampt^f/+ (control) and Nampt^f/f;Prx1-Cre (Nampt^ΔPrx1) mice at P28. **b** H&E staining performed on histological sections of knee, tibia, and ankle of Nampt^ΔPrx1 mice to illustrate the lack of mineralized secondary ossification center (*n* = 3 mice/group). **c** Representative pictures of H&E staining of undemineralized histological sections of tibia (left) and quantification of osteoblast numbers at the endocortical surfaces (right) (*n* = 3 mice/group). Black arrows indicate osteoblasts

covering the endocortical surfaces. BM=bone marrow. **d** Representative pictures of double calcein labels at the endocortical surfaces indicated by the white arrows (left) and quantification of mineral apposition rate (MAR) (right). Dotted white line marks the bone surface. **e** Representative pictures of male mice. **f** Whole body weight, and (**g**) DXA BMD of 1 mo-old male (*n* = 5 Nampt^ΔOsx1 and 12 Osx1-cre mice/group) and female mice (*n* = 8 Nampt^ΔOsx1 and 11 Osx1-cre mice/group). Lines represent mean ± SD, *p* values by two-sided Student t-test. Source data are provided as a Source Data file.

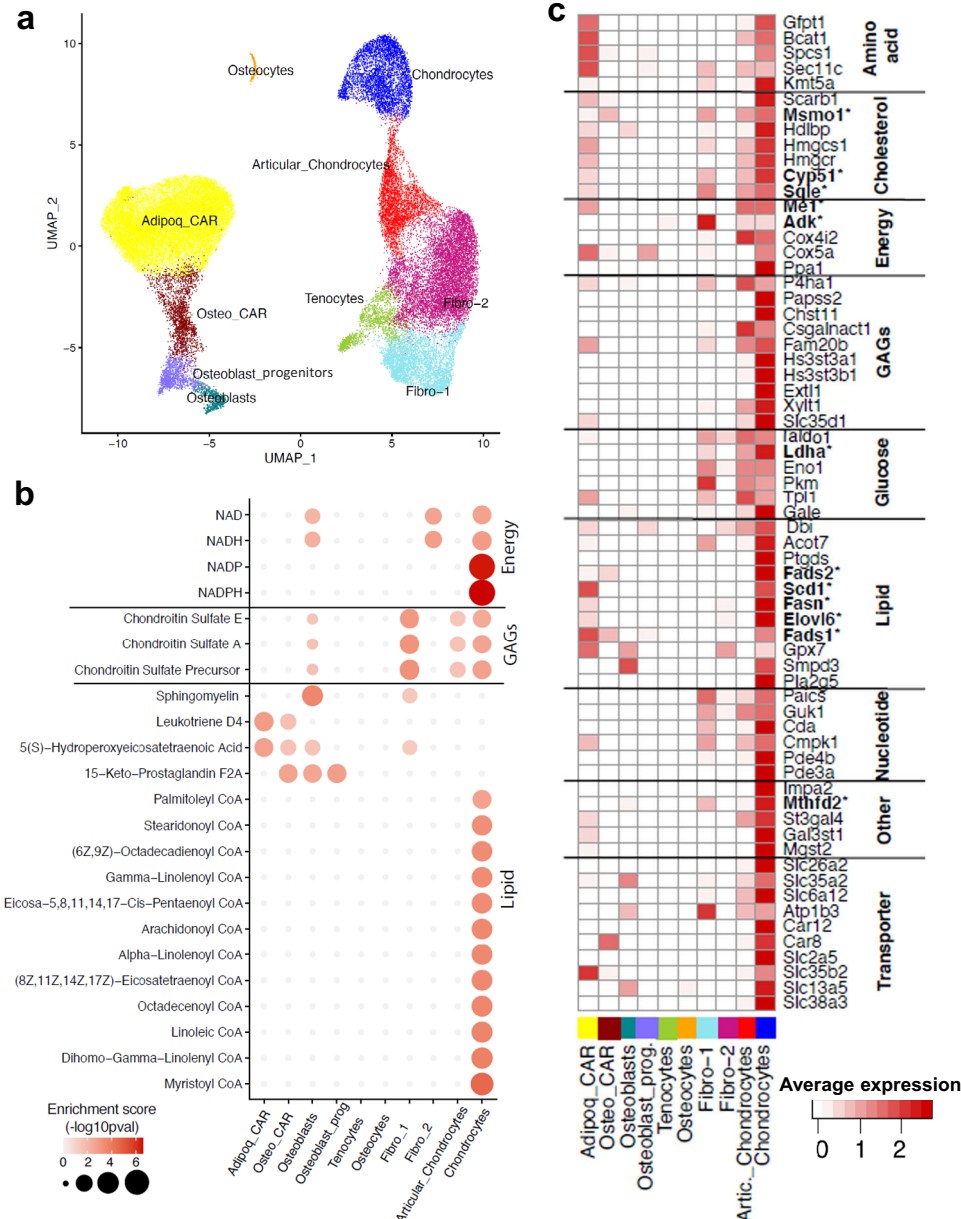

**Fig. 6 | Chondrocytes have enhanced redox metabolism. a** Single-cell RNA-seq analysis, UMAP-based visualization of major classes of non-hematopoietic cells (bone and bone marrow fractions) based on two published single cell RNA sequencing datasets, annotated post hoc (see Supplementary Fig. 2 for cell specific markers of individual cell types) and colored by clustering. **b** Dot plots of selected reporter metabolites of individual cell types using reporter metabolite analysis test[26]. **c** Expression (row-wide Z score of normalized expression level; single cell RNA sequencing) of metabolic related genes (rows) in the cells of each cluster (columns). Depicted in bold with an asterisk are the enzymes that require NAD/NADH or NADP/NADPH redox reactions.

secreted to the medium confirmed that cells in hypoxia were using glycolysis to produce energy (Fig. 7f, g), and that FK866 decreased the levels of lactate after 2 d but not 1 d of treatment. The similarities between NAD⁺ and ATP depletion in normoxia and hypoxia do not

support the idea that hypoxia alone is responsible for the high sensitivity of growth plate chondrocytes to Nampt inhibition. Most likely, the low nutrient supply is a major contributor to the dependence of chondrocytes on the NAD salvage pathway.

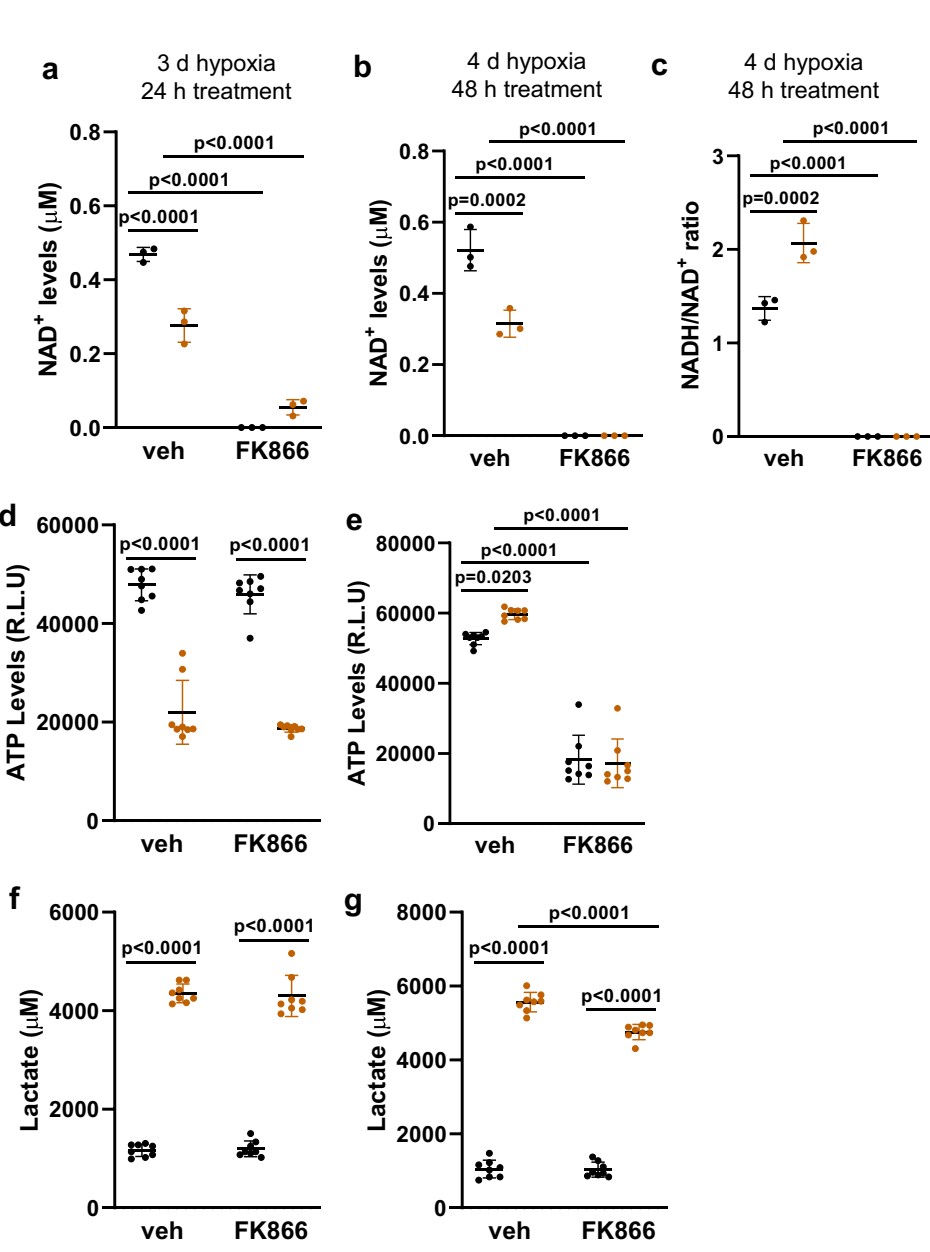

**Fig. 7 | Nampt inhibition has a similar effect on ATP production in chondrocytes cultured in normoxia versus hypoxia.** Primary chondrocytes from wild-type C57BL/6 mice were cultured in normoxia (20% O₂) or hypoxia (1% O₂) for a total of 3 days and treated with vehicle or FK866 for the last 24 h (**a**, **d**, **f**), or for a total of 4 days and treated for the last 48 h (**b**, **c**, **e**, **g**). Measurements of intracellular

NAD⁺ (*n* = 3 wells) (**a**, **b**), NADH/NAD ratio (*n* = 3 wells) (**c**), intracellular ATP (*n* = 8 wells) (**d**, **e**), and lactate in the medium (*n* = 8 wells) (**f**, **g**). Graphs depict representative experiments. Each dot represents replicate wells. Each experiment was repeated twice. Lines represent mean ± SD, *p* values by Student t-test. Source data are provided as a Source Data file.

## Deletion of *Nampt* significantly impacts the transcriptomic profile of growth plate chondrocytes

Next, we performed scRNA-seq of cells isolated from the growth plate of P2 mice born from dams that were fed NR during gestation. After quality control, we profiled 13,965 high-quality cells (6788 and 7177 cells derived from control and *Nampt*[ΔPrx1], respectively) with an average of over 47,000 unique molecular identifiers (UMIs) per cell and an average of approximately 3000 genes per cell. An initial UMAP plot and cluster analysis, after harmonization of cells from mice of the two genotypes, revealed 11 clusters (Supplementary Fig. 7) according to the expression of cluster-specific markers (Supplementary Fig. 8). Among the cells analyzed, 6693 cells were chondrocytes, 4449 mesenchymal

cells (Fibro-1, Fibro-2, and Fibro-3), 1303 were tenocytes, and 1520 were mural cells (Pericytes I, Pericytes II, and endothelial cells). Because each of the pericyte and endothelial cell clusters contained a small number of cells, we excluded these and considered the 8 major clusters for further analysis (Fig. 8a). Chondrocyte subpopulations from both control and *Nampt*[ΔPrx1] mice included chondrocytes I, chondrocytes II, articular chondrocytes, and proliferating chondrocytes. These clusters all expressed markers such as *Col2a1, Acan* and *Sox9*, while *Prg4* was expressed in articular chondrocytes (Fig. 8b). Other clusters included mesenchymal precursor cells designated fibro-1, fibro-2 [identified based on previous studies[24,25]], as well as fibro-3 and tenocytes (including ligamentocytes) (Fig. 8a). Specifically, fibro-1

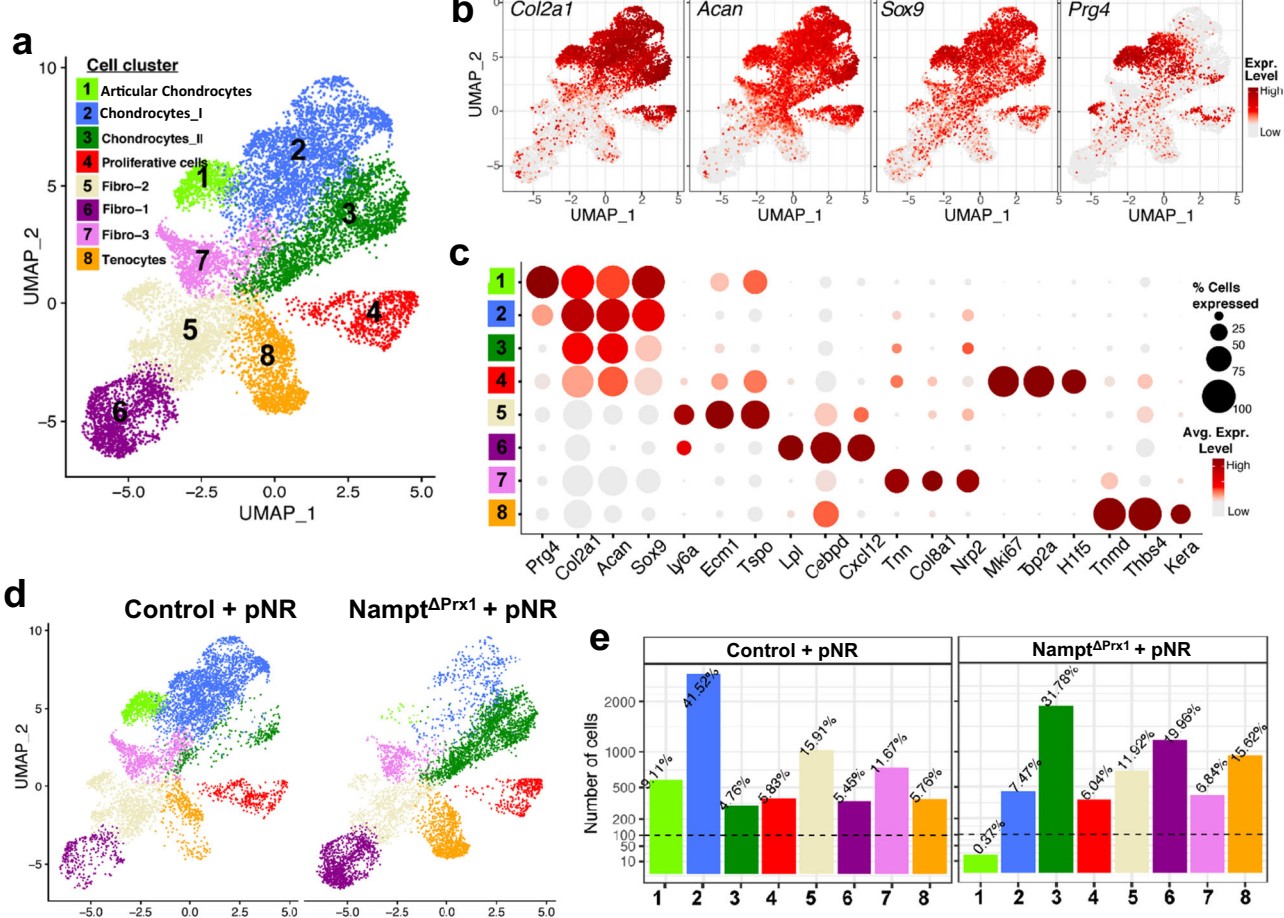

**Fig. 8 | Nampt deletion greatly impacts gene expression in chondrocytes.** NR was administered to dams during pregnancy. Single cell RNA-seq analysis of cells extracted from the growth plates of P2 Nampt^ΔPrx1 and littermate control mice born from dams receiving NR during pregnancy. **a** UMAP plot of chondrocytes and other mesenchymal lineage cells obtained from mice of both genotypes. **b** Gene expression plot of key known chondrocytes markers of individual cells on UMAP coordinate. **c** Dot plot of the distribution of top markers gene in Clusters 1–8. **d** UMAP plots of the different cell types described in panel a obtained from control versus Nampt^ΔPrx1 mice; pre-natal NR (pNR). **e** Bar plots depicting the number and percentage of cells within the cell clusters shown in (**d**).

expressed common stem cell markers *Ly6a, Ecm1,* and *Tspo*; fibro-2 expressed *Lpl, Cebpd,* and *Cxcl12*; fibro-3 expressed *Tnn, Col8a1,* and *Nrp2*; tenocytes expressed *Tnmd, Thbs4,* and *Kera* (Fig. 8c). Deletion of *Nampt* led to a drastic decrease in the number of articular chondrocytes as well as *Prg4*-expressing fibro-2; for non-*Prg4*-expressing chondrocytes, there was a shift from cluster I to cluster II (Fig. 8d, e). Interestingly, the percentages of proliferative chondrocytes and other mesenchymal cells were similar in control versus *Nampt*^ΔPrx1 mice (Fig. 8e). Deletion of *Nampt* also increased the relative numbers of fibro-1 and tenocytes.

### Deletion of *Nampt* suppresses multiple metabolic pathways in chondrocytes

Because a major consequence of *Nampt* deletion was the shift in cell number between chondrocytes I and II, we focused our analysis on the comparison between these two clusters. Differential gene expression analysis (adjusted $p < 10^{e-10}$ and $\log_2$ average fold change <0.5) identified 50 upregulated genes and 3642 downregulated genes when comparing chondrocytes II to chondrocytes I (Fig. 9a and Supplementary Data 1). This strongly indicated the abnormal transcriptional program of chondrocytes from Nampt^ΔPrx1 mice. Specifically, the expression of many genes encoding collagens (*Col9a3, Col11a1, Col9a1, Col9a2*) and other extracellular matrix (ECM) proteins (*Epyc, Dcn, Snorc, Ucma*) were suppressed in chondrocytes II (Fig. 9a). In contrast,

the expression of genes involved in stress responses such as *Hmga2, Spry2, Ptgs2* (*Cox2*), and *Neat1* as well as genes previously associated with osteoarthritic chondrocytes such as *Postn, Tnfai6, Tmsb4x* were upregulated in chondrocytes II (Fig. 9a and Supplementary Fig. 9). RNA in situ hybridization confirmed the elevated expression of *Ptgs2* and *Pim3* in the growth plates from Nampt^ΔPrx1 hindlimbs (Fig. 9b and Supplementary Fig. 10). Interestingly, the expression of *Ptgs2* was localized to the cells on the bone surfaces and adjacent synovium, perichondrium, and menisci. In control mice, *Ptgs2* was mainly restricted to the proliferative and hypertrophic zone (Supplementary Fig. 10). The localization of cell expressing *Ptgs2* was reminiscent of the localization of TUNEL positive cells in Nampt^ΔPrx1 mice of the same experimental design (Fig. 3). *Pim3*, which was upregulated in a sub-population of chondrocytes II distinct from those expressing *Ptgs2* (Supplementary Fig. 9), was elevated in multiple growth plate zones (Fig. 9b and Supplementary Fig. 10).

Gene Ontology (GO) enrichment analysis of differentially expressed genes revealed that cellular components and biological processes related to mitochondria activity, translation, protein secretion, and carbohydrate metabolic processes were downregulated in chondrocytes II. On the other hand, biological processes such as angiogenesis, cell adhesion, and cell death were upregulated in chondrocytes II (Fig. 9c). Focusing on cellular metabolism, reporter metabolite analysis revealed that metabolic hotspots for amino acids

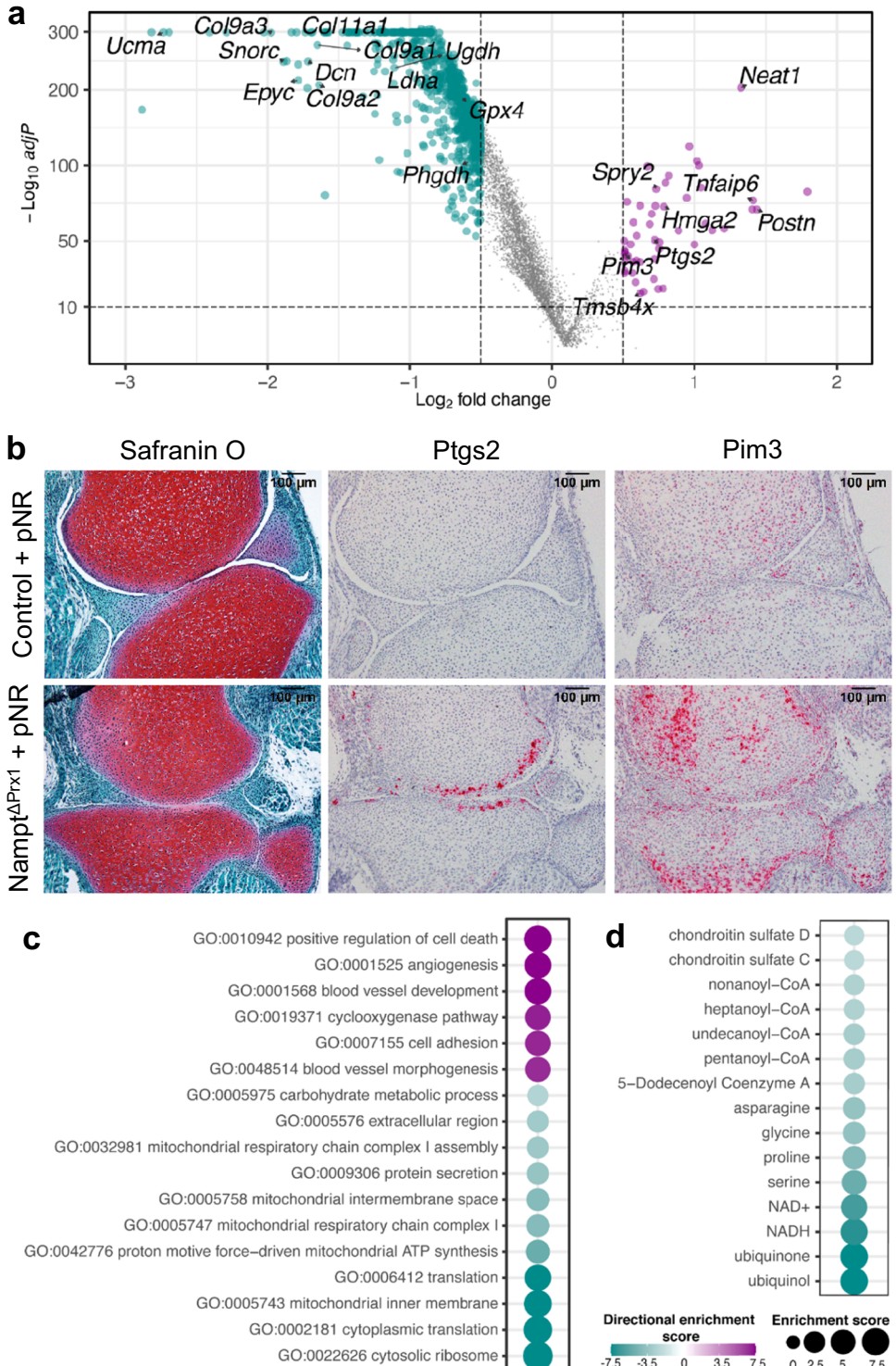

**Fig. 9 | Nampt deletion severely impacts cell metabolism. a** Volcano plot of the differential gene expression results between chondrocytes II and chondrocytes I, described in Fig. 8, using two-sided MAST statistical test[69] with false discovery rate adjusted p-value. Some key genes are labeled on the plot. Genes downregulated in chondrocytes II are in green and upregulated are in purple. **b** Serial sections from P2 Nampt$^{f/+}$ (control) and Nampt$^{f/f}$;Prx1-Cre (Nampt$^{\Delta Prx1}$) knees (representative of 2–3 pups per genotype) showing Safranin O staining for proteoglycans (left), in situ hybridization for *Ptgs2* (center) and *Pim3* (right); pre-natal NR (pNR). **c, d** Dot plots of key results of functional enrichment analysis based on the differential gene expression between chondrocytes II and chondrocytes I. Processes enriched in chondrocytes II are in purple and processes suppressed are in green based on gene ontology (GO) (**c**), and reporter metabolite (**d**).

such as serine, proline, glycine and asparagine were reduced in chondrocytes II (Fig. 9d). Proline and glycine constitute the backbone of collagen molecules, therefore a reduction in metabolism of these amino acids in chondrocytes II is in agreement with the decreased expression of the major collagen components of the ECM. Furthermore, processes that involve redox reactions dependent on NAD$^+$/NADH were decreased in chondrocytes II, in line with the increase in this abnormal chondrocyte population in *Nampt* deficient mice

(Fig. 9d). The expression of key enzymes implicated in carbohydrate metabolism such as *Ldha*, *Phgdh*, and *Ugdh* which mediate anaerobic glycolysis, serine synthesis, and biosynthesis of GAGs, respectively, was suppressed in chondrocytes II (Fig. 9a and Supplementary Data 1). Reporter metabolite analysis also revealed that ubiquinone/ubiquinol metabolism was reduced in chondrocytes II (Fig. 9d). Ubiquinone is necessary for mitochondrial energy production, is a prominent source of ROS, and has antioxidant properties in its reduced form (ubiquinol)[36].

### Chondrocytes lacking *Nampt* exhibit metabolic defects

To examine the functional relevance of some of the key findings of scRNA-seq analyses, we cultured cells from the growth plates of control and *Nampt*[ΔPrx1] mice. As expected, cells lacking *Nampt* had decreased levels of NAD[+] (Supplementary Fig. 11a). We next examined the rate of glycolysis, as this is the major energy source in chondrocytes. As indicated by the extracellular acidification rate (ECAR) measured using SeaHorse, cells from *Nampt*[ΔPrx1] mice exhibited decreased glycolysis (Supplementary Fig. 11b). Accordingly, the levels of ATP were lower in cells lacking *Nampt* (Supplementary Fig. 11c). We also found that the levels of ROS were increased in these cells (Supplementary Fig. 11d), perhaps due to the altered ubiquinone/ubiquinol and other metabolic processes. Because NAD[+] is a critical cofactor for proteins such as sirtuins and PARPs, we examined by Western blot the levels of lysine acetylation and the levels of PAR, indicators of Sirtuin and PARP activity, respectively. A modest decrease in PAR and increase in lysine acetylation were seen in cells from *Nampt*[ΔPrx1] mice (Supplementary Fig. 11e). We also examined whether expression of the multiple sirtuins or Parps was altered in chondrocytes II versus chondrocytes I. Of the 7 sirtuin family members, expression of *Sirt1* and *Sirt2* was slightly decreased in chondrocytes II (Supplementary Data 1). A small decrease in Sirt1 was confirmed at the protein level in cultured chondrocytes from *Nampt*[ΔPrx1] mice (Supplementary Fig. 11e). Of the 17 Parp family members, expression of *Parp2* and *Parp8* was decreased in chondrocytes II (Supplementary Data 1). Some of the Parp family members, including Parp2 have been implicated in DNA damage repair[37]. Excessive ROS and decreased Parp activity can lead to accumulation of DNA damage. Therefore, we examined the protein levels of γ-H2AX, an indicator of DNA damage, but found no differences between cells of control and *Nampt*[ΔPrx1] mice (Supplementary Fig. 11e). Nonetheless, because the culture conditions do not replicate the limited nutrient supply experienced by cells in the growth plate, it is very likely that the changes observed in cultured cells are attenuated relative to the in vivo condition. Together with the scRNA-seq analyses, these findings indicate that *Nampt* deletion led to numerous metabolic derangements in chondrocytes that caused cellular stress and rapid cell death.

## Discussion

NAD is indispensable for life and deficiencies in NAD cause diseases such as pellagra and acute kidney insufficiency, as well as congenital defects in multiple organs including the heart, vertebrae, kidneys, and limbs[12]. Despite the requirement of NAD for multiple essential processes, the contribution of the recycling pathway to maintain cellular NAD varies among the different cell types. Here, we show that the NAD salvage pathway is indispensable for endochondral bone formation during embryonic development and post-natal bone growth. Although NAD can be synthesized de novo from tryptophan, it has been previously shown that mouse embryos lacking *Haao* or *Kynu*, enzymes in the de novo NAD pathway, do not exhibit developmental defects unless the dams are placed on a niacin (NA, NAM, and NR)-deficient diet[12]. This is in line with the assumption that most of NAD derives from pathways which require the uptake of other NAD precursors from the diet. During development, embryos receive niacin from the dam and produce their own. We show that the skeletal defects caused by *Nampt*

deletion occur when mice are fed a normal diet. While NA is an abundant NAD precursor in circulation, it is unknown whether chondrocytes can convert NA into NAD via the Preiss–Handler pathway. In any case, our findings demonstrate that the cell intrinsic conversion of NAM into NMN in the salvage pathway is indispensable for proper endochondral bone development in mice and that the de novo and Preiss–Handler pathways of NAD synthesis, if present, are not sufficient to provide the NAD required for this process.

We found that NR supplementation during pregnancy prevented the development of a very severe phenotype caused by genetic *Nampt* deletion in the embryos. NR is a dietary precursor that can be used by the salvage pathway to produce NAD. Intracellularly, NR is converted to NMN by NRK1 and NRK2. NMN, in turn, is converted to NAD[+] in the last step of the salvage pathway, bypassing the need for Nampt. Previous studies have highlighted the importance of an adequate maternal intake of NAD precursors to avoid defects in multiple tissues namely the heart, kidney, and skeleton[19]. Our current findings elucidate that the critical roles of NAD during skeletal development are exerted directly and not secondary to the dysfunction of other tissues. Our findings also raise the exciting possibility that an elevated intake of NR during pregnancy might mitigate unforeseen developmental defects arising from genetic deficiencies in NAD synthetic pathways[18]. Although we did not administer NR to *Nampt*[ΔPrx1] mice after birth, this most likely would have been effective in preventing the post-birth defects that occur in these mice. This premise is further supported by evidence that NR administration post-birth alleviates defects caused by specific deletion of Nampt in muscle or liver[38,39]. NMN administration also alleviates defects caused by deletion of Nampt in multiple tissues[9,40].

The resting zone of the growth plate contains mesenchymal precursors which differentiate into chondrocytes that proliferate and hypertrophy[41,42]. Our findings indicate that the survival of cells present in the resting zone of the growth plate was severely compromised in the absence of Nampt. Likewise, the survival of chondrocytes in the growth plate after birth was compromised by *Nampt* deletion. Remarkably, chondrocytes forming the articular surfaces of hindlimb joints were especially sensitive to *Nampt* deletion, displaying TUNEL positivity and upregulation of the inflammatory mediator *Ptgs2* (Cox-2) within days of NR withdrawal. These same markers characterize articular cartilage degeneration during osteoarthritis[43], suggesting a possible role for the NAD salvage pathway in articular chondrocyte homeostasis.

Hypoxia-Inducible Factor (Hif) 1α promotes chondrocyte survival in the hypoxic environment by impairing mitochondrial respiration[35,44]. Indeed, conditional ablation of *Hif-1α* in growth plate chondrocytes and the limb bud mesenchyme results in defective skeletal development associated with massive cell death in the inner chondrocyte layer of the growth plate[44]. We examined in vitro whether hypoxic cells could be more sensitive to a decrease in the NAD salvage pathway but found no evidence that this was the case. It is possible that inhibition of glycolysis, the major source of energy in chondrocytes and a process dependent on the NAD[+]/NADH redox pair, is responsible for the effects caused by *Nampt* deletion. In line with the dependence on glucose metabolism, deletion of *Glut1*, the major glucose transporter in chondrocyte, causes shortening of the limbs[45]. However, in *Glut1*-deficient mice no chondrocyte death was detected in the growth plate; rather, defects were attributed to decreased chondrocyte proliferation and hypertrophy. Redox reactions are also essential to fuel lipid and cholesterol synthesis. We found that the expression of multiple enzymes required for these metabolic processes, including the ones that use NADH and NADPH as cofactors, are elevated in chondrocytes when compared to all other mesenchymal cells in bone. In addition, deletion of *Nampt* decreased the expression of genes implicated in lipid metabolism. Genetic disorders of cholesterol synthesis are characterized by multiple congenital abnormalities, including

significant skeletal defects[46]. Studies in rodents have further elucidated that intracellular cholesterol and lipid biosynthesis in chondrocytes are crucial for normal endochondral ossification, joint formation, and bone growth[47,48]. Similar to our findings with *Nampt* deletion, disruption of cholesterol synthesis in chondrocytes causes cell death in the developing growth plate[48]. Together, this evidence supports the idea that alterations in cholesterol biosynthesis could contribute to the defects caused by *Nampt* deletion in the growth plate. In line with this idea are the findings that altered lipid homeostasis plays a critical role in mediating cell death caused by Nampt inhibition in acute myeloid leukemia stem cells[49].

In contrast to the severe impact on bones formed by endochondral ossification, bones formed by intramembranous ossification, such as bones in the skull, were not affected by *Nampt* deletion. In addition, endocortical osteoblasts and bone formation were not overtly impacted in the absence of Nampt during early development. However, skull defects along with other skeletal malformations in embryos have been reported when wild-type pregnant mice are placed on diets with restricted NAD precursors[19]. In developing mouse limbs and cranial bone, the appearance of osteoblasts coincides with the invasion of blood vessels near sites of initiation of the primary ossification center[13]. These observations suggest that the supply of NAD precursors from the circulation or from neighboring cells is sufficient to sustain normal osteoblast differentiation during intramembranous ossification, and that the dependency on the NAD salvage pathway is not critical as in the case of endochondral ossification. Our findings that *Nampt* ablation using Osx1-Cre did not cause any overt bone defects further demonstrate the distinct reliance on the NAD salvage pathway of chondrocytes versus osteoblasts, two cell types that originate from a common mesenchymal precursor and are indispensable for bone development. We have previously shown that mice lacking one allele of *Nampt* in Prx1-Cre targeted cells have a normal skeleton up to 2 month-of-age[11]. Yet, these mice exhibit lower bone mass at 8 months when compared to littermate controls, indicating that the NAD salvage pathway in mesenchymal lineage cells also contributes to bone formation. Future studies are warranted do elucidate the role of the NAD salvage pathway in distinct bone populations.

Our studies indicate that the dependency of growth plate chondrocytes on the NAD salvage pathway is due to the limited supply of nutrients. In addition, the comparison of the metabolic hotspots among different bone cell types revealed that chondrocytes are particularly enriched in metabolic processes dependent on NAD. The unbiased gene expression analysis using scRNA-seq elucidated the extent and severity of changes that occur in growth plate chondrocytes deficient in Nampt. Namely, we observed an overall decrease in gene transcription, and in particular, in genes involved in mitochondria metabolism, glycolysis, and lipid metabolism. These results are consistent with studies performed using bulk RNA-seq in tissues such as skeletal muscle, liver, and projection neurons, which show that Nampt deletion results in an overall downregulation of genes involved in metabolic processes[39,50,51]. In contrast, our analysis indicated that Nampt deletion in chondrocytes led to a severe reduction in translation and protein secretion, processes that were not impacted in other tissues[39,50,51].

In multiple tissues the detrimental effects of *Nampt* deletion have been attributed to inhibition of Sirt1 activity[10]. However, mice with deletion of Sirt1 in mesenchymal cells targeted by Prx1-Cre or Col2a1-Cre develop normally[52,53], indicating that Sirt1 in chondrocytes has no major functions during skeletal development. Likewise, mice with individual deletion of Sirt3, Sirt6, Parp1, or Parp2 do not exhibit developmental limb defects[54–57]. While our studies in chondrocytes cultured from *Nampt*[ΔPrx1] mice did not reveal major changes in markers of Sirtuin and PARP activity, it remains possible that the combined inhibition of all Sirtuins and PARPs contributes to the cell death seen in Nampt[ΔPrx1] mice. Because NAD$^+$ is required as both a substrate and

cofactor for numerous metabolic enzymes, a range of metabolic disturbances occurring simultaneously likely contributes to chondrocyte death in *Nampt*[ΔPrx1] mice.

A plethora of evidence gathered during the last decade in lower organisms and rodents have shown that restoration of NAD levels using metabolic precursors leads to multiple benefits to health in old age[9]. These have raised expectations about the use of NAD-based therapeutics in combating diseases of aging, underlined by the multiple clinical trials to test the efficacy of these drugs in humans[58–62]. Our studies demonstrate that administration of NR during pregnancy in mice can prevent severe developmental defects caused by genetic inhibition of NAD production. In view of the multiple clinical syndromes already linked to defective NAD production, our findings raise hope of their attenuation and perhaps prevention.

## Methods
### Animal experimentation
Animal studies were performed as pre-approved by the University of Arkansas for Medical Sciences (UAMS) Institutional Animal Care and Use Committee. Mice with conditional deletion of *Nampt* in the mesenchymal lineage were generated by a two-step breeding strategy. Male hemizygous Prx1-cre transgenic mice (B6.Cg-Tg(Prrx1-cre)1Cjt/J; Jackson Laboratories, stock # 5584) were crossed with *Nampt* floxed ($^{f/f}$) mice (C57BL/6 genetic background) (provided by Shin-ichiro Imai, Washington University School of Medicine) to generate mice hemizygous for the *Nampt* floxed allele with and without the Cre allele, *Nampt*[f/+;ΔPrx1] and *Nampt*[f/+], respectively. Male *Nampt*[f/+;ΔPrx1] and female *Nampt*[f/+] mice were intercrossed to generate *Nampt*[f/f] and *Nampt*[ΔPrx1] mice. Mice with conditional deletion of *Nampt* in the osteoblast lineage were generated using a similar strategy. Specifically, hemizygous Osx1-Cre transgenic mice[63] (B6.Cg-Tg (Sp7-tTA,tetO-EGFP/cre) 1Amc/J; Jackson Laboratories, stock # 6361) were crossed with *Nampt*[f/f] mice, to generate mice heterozygous for the *Nampt* floxed allele with and without the Cre allele. These mice were intercrossed to generate the experimental Osx1-Cre (control) and *Nampt*[ΔOsx1] mice. Offspring were genotyped by PCR using the following primer sequences: Nampt-flox primer #1 5'TTC CAG GCT ATT CTG TTC CAG 3' and primer #2 5' TCT GGC TCT GTG TAC TGC TGA 3'. Offspring from all genotypes were tail-clipped for DNA extraction at the time of sacrifice or at weaning (21 days) and then group-housed with same sex littermates. To allow for quantification of bone formation rates, mice were injected with calcein (35 mg/kg body weight) 7 and 3 days before euthanasia at 1 month of age. In the NR rescue experiments, mice were treated with 1000 mg/kg NR (Tru Niagen, ChromaDex) administered *ad libitum* in the drinking water in light-protected bottles. The NR water solution was filtered and replaced daily during conception and pregnancy. Rescued *Nampt*[ΔPrx1] pups and control littermates were monitored daily from birth until the experimental endpoint at P2, P7, or P28. Male and female mice were used. For all experiments, mice were maintained with a constant temperature of 23 °C, a 12-h light/dark cycle, and had access to chow (LabDiet #5V5R) and water *ad libitum*. Body weight measurements were performed before euthanasia by either decapitation (neonates up to 10 days of age) or CO$_2$ inhalation with cervical dislocation (mice beyond 10 days old).

### Whole-mount skeletal staining
Both male and female newborn mice were used without distinction in all analyses. After removing the skin and internal organs, mice were fixed overnight in 95% ethanol and then stained overnight with 0.015% Alcian blue (Sigma Aldrich, A3157), 20% Glacial acetic acid, and 80% ethanol. Next, specimen were transferred to 95% Ethanol for 3 h, and cleaned up with 2% KOH for 24 h. The skeletons were stained overnight with a solution containing 0.005% Alizarin Red (Sigma Aldrich, A5533), in 1% KOH. The stained skeletons were cleared in 1% KOH/ 20% glycerol solution for 2 days, and stored in 1:1 mixture of glycerol/ethanol 95%.

## Histology and immunohistochemistry

For paraffin sectioning limbs and sterna were dissected out and fixed in freshly made 4% Paraformaldehyde overnight at room temperature. Bones were decalcified for 1 week with 14% EDTA (pH 7.2) followed by paraffin embedding. Sections were stained with H&E or Safranin O/Fast Green. TUNEL assay was performed using an in situ Calbiochem® Fluorescein-Fragel™ DNA Fragmentation Detection kit (Millipore-Sigma). Following antigen retrieval using either 10 min 0.1% pronase and 30 min 0.1% hyaluronidase (type II collagen) or 10 mM Tris-EDTA (pH 9.0) at 65 °C for 2 h (Nampt), immunostaining was performed using a mouse IgG1 for type II collagen (clone II-II6B3; Developmental Studies Hybridoma Bank, University of Iowa, 1:1000) or a rabbit monoclonal antibody for Nampt (clone EPR21980; Abcam ab236874, 1:3000) or isotype control (clone EPR25A, Abcam ab172730, 1:9000). Primary antibody detection was performed using Vectastain Elite ABC-HRP Mouse (PK6102) or Rabbit (PK6101) IgG Detection Kits (Vector Labs). For undecalcified sections, freshly dissected femurs were fixed for 24 h in 10% Millonig's formalin with 5% sucrose, transferred to ethanol and embedded undecalcified in methyl methacrylate. Calcein labels and osteoblasts were quantified on 5 µm thick longitudinal/sagittal nondecalcified sections using the OsteoMeasure Analysis System (OsteoMetrics, Inc. Atlanta, GA).

## Bone mineral density and micro-CT imaging

Dual-energy X-ray absorptiometry was performed on sedated mice (2% isoflurane) using a PIXImus densitometer (GE Lunar) and data analyzed as described previously[64]. Whole body (excluding the head) or the entire right femur scans were used to determine BMD. For micro-CT analysis, femurs were cleaned of adherent tissue, fixed in Millonig's (Leica Microsystems), and stored in 100% ethanol, loaded into 10-mm diameter scanning tubes, and imaged (micro-CT40; Scanco Medical, Brüttiselen, Switzerland), as described previously[65]. Scans were performed at medium resolution (12 µm isotropic voxel size) for quantitative determinations and integrated into 3-D voxel images (1024 × 1024 pixel matrices for each individual planar stack). A Gaussian filter (sigma = 0.8, support = 1) was applied to all analyzed scans. Key parameters were X-ray tube potential = 55 kVp, X-ray intensity = 145 µA, integration time = 200 ms, and threshold = 200 mg/cm$^3$.

## Primary chondrocyte cultures

The rib cages from 3–5 day-old C56BL/6 mice were dissected free of soft tissues and digested with 2 mg/ml of pronase (Sigma Aldrich, P69111) for 30 min followed by 3 mg/ml of collagenase D (Sigma Aldrich, P6885) for 1.5 h to further remove attached soft tissues. To dissociate the chondrocytes, the rib cages were further digested with 3 mg/ml of collagenase D for 2 h. Chondrocytes were then filtered through 40 µm cell strainer (Falcon™ 352340), collected by centrifugation, and cultured in DMEM with 10% FBS. Primary chondrocytes were also isolated from control and Nampt$^{ΔPrx1}$ mice born from dams that were administered NR during pregnancy. Within 6 h after birth, forelimbs and hind limbs were isolated with micro scissors from. Soft tissues surrounding the longs bones and cartilages were carefully cleaned with micro forceps without damaging the growth plate. Remaining soft tissues were further removed with enzymatic digestion using 2 mg/ml pronase for 30 min followed by Liberase TM (Sigma/Roche; 2.5 Wunch unit/mL in 1% FBS) for 1 h. Growth plates were then dissected with a surgical blade and washed 3 times in PBS (Gibco™ 10010072). Chondrocytes were dissociated with Liberase TM (Sigma/Roche; 2.5 Wunch unit/mL in 1% FBS) for 1 h. Cells were filtered through 40 µm cell strainer (Falcon™ 352340), collected by centrifugation, and cultured in DMEM supplemented with 10% FBS and 1% penicillin/streptomycin at 1% O$_2$ concentration in a hypoxia chamber (Coy Laboratory Products Inc).

## NAD, NADH, ATP, and Lactate

NAD$^+$ and NADH levels were measured using the EnzyFluo™ NAD$^+$/NADH Assay kit, according to the manufacturer's instructions (Bioassay Systems). Briefly, primary chondrocytes were plated in a 96-well black-wall tissue plate and cultured in 1% or 20% O$_2$ for up to 4 days. After 1 or 2 d cultures with or without FK866, NAD and NADH extraction buffers were added to the respective wells, following the manufactured instructions. The fluorescent signal ($λ_{ex/em}$ = 530/585 nm) was quantified in a Cytation 5 reader.

Intracellular ATP levels were measured by a luciferin-luciferase based assay using CellTiter-Glo® Luminescent Cell Viability Assay (G7570, Promega). Primary chondrocytes were seeded in a white opaque-bottom 96-well plate (8 × 10$^4$ cells/cm$^2$) and cultured in 1% or 20% O$_2$ as above. At the end of treatment, cell culture media was replaced with 100 µl of assay reagent (CellTiter-Glo Buffer + CellTiter-Glo Substrate). Contents were mixed for 2 min on an orbital shaker to promote cell lysis, followed by a 10 min incubation at room temperature. The luminescence signal was monitored in a Cytation 5 reader.

Lactate assay was performed using Glycolysis Cell-Based Assay (600450, Cayman). Primary chondrocytes were seeded and treated as described above. After treatment, the media was collected to analyze the lactate amount according to the instructions provided in the kit. The absorbance signal (490 nm) was recorded in a Cytation 5 reader.

## Extracellular acidification rate (ECAR)

Real-time measurements of ECAR were determined using Seahorse XFp setup and Seahorse XFp Glycolysis Stress Test Kit (Agilent 103020-100) according to the manufacturer's protocol. Briefly, chondrocytes seeded at a density of 4.0 × 10$^4$/well in Seahorse XF96 Cell Culture Microplates in high glucose DMEM medium (Gibco™ 11995073) were washed twice and replaced with Agilent Seahorse XF Base medium supplemented with 2mM L-glutamine, 1 h before the assay. The basal ECAR rate, defined as the number of protons exported by cells into the assay medium over time was recorded 3 times over a period of 20 min and expressed as pmol/min. Cells were stained with Hoechst 33342 nuclear stain as detailed above. Raw ECAR values were normalized to the cell number count of each well.

## Reactive oxygen species (ROS)

Primary chondrocytes plated in a 96-well black-wall tissue plate were incubated for 15 min at 37 °C with 5 µM CM-H$_2$DCFDA probe (C6827, Invitrogen). Cells were washed twice with 1× PBS, and the fluorescent signal ($λ_{ex/em}$ = 492/517 nm) was measured in a Cytation™ 5 microplate reader. DCFDA fluorescence was normalized to the cell number count of each well after staining with Hoechst 33342 (Invitrogen™ H3570) nuclear stain. Hoechst fluorescence was analyzed with Cytation 5 Cell Imaging Multi-Mode Reader (BioTek Instruments Inc, Winooski, VT, USA).

## Quantitative RT-PCR (qRT-PCR)

Total RNA from femoral growth plates was extracted with TRIzol reagent (Invitrogen) and reverse-transcribed using the High Capacity Reverse Transcription Kit (Applied Biosystems) according to the manufacturer's instructions. TaqMan quantitative real-time PCR was performed in a QuantStudio3 (Applied Biosystems) using the following primers (Applied Biosystems): *Col2a1* (Mm01309565_m1); *Col10a1* (Mm00487041_m1); *Mmp-13* (Mm00439491_m1); *Vegfa* (Mm00437306_m1); *Acan* (Mm00545794_m1); Runx2 (Mm00501584_m1). Target gene expression was calculated by normalizing to the housekeeping gene ribosomal protein S2, *Mrps2* (Mm00475528_m1) using the ΔCt method[66].

## Western blot analysis

Cultured cells were washed twice with ice-cold PBS and lysed with a buffer containing 20 mM Tris-HCL, 150 mM NaCl, 1% Triton X-100,

protease inhibitor mixture, and phosphatase inhibitor cocktail (Sigma-Aldrich) on ice for 30 min. The cell lysates were centrifuged at 11,200 × g for 10 min at 4 °C and the supernatants were collected in new tubes. The protein concentration of cell lysates was determined using a DC Protein Assay kit (Bio-Rad). The extracted protein (40 µg per sample) was subjected to 8 to 12% SDS-PAGE gels and transferred electrophoretically onto polyvinyl difluoride membranes (Merck Millipore). The membranes were blocked in 5% fat-free milk/Tris-buffered saline for 120 min and incubated with a primary antibody followed by a secondary antibody conjugated with horseradish peroxidase. The following primary antibodies were used to detect their corresponding protein levels: rabbit monoclonal antibody against Nampt (clone EPR21980, Abcam, ab236874, 1:1000); rabbit polyclonal antibody aagainst Acetylated-Lysine (Cell Signaling, #9411, 1:1000); mouse monoclonal antibodies against PAR (clone 10H, Enzo Life Sciences, ALX-804-220-R100, 1:1000), Sirt1 (clone 1F3, Cell Signaling, #8469, 1:1000), phospho-Histone H2A.X (clone JBW301, Sigma-Aldrich, #05-636, 1:1000), and β-actin (clone ACTBD11B7, Santa Cruz Biotechnology, sc-81178, 1:2000). The membranes were subjected to western blot analysis with ECL reagents (Millipore) and imaged with a VersaDoc™ imaging system (Bio-Rad). All blots were derived from the same experiment and were processed in parallel. Unprocessed scans of the Western blots are shown in Supplementary Fig. 12.

### In situ hybridization

Sample processing for ISH using RNAscope™ technology was performed under RNase-free conditions. Hind limbs from P2 control or Nampt[ΔPrx1] mice were quickly excised and fixed in Millonig's buffer for 24 h at 4 °C, followed by decalcification in 15% EDTA in PBS (pH 7.4) for 1 week at 4 °C. Dehydrated hind limbs were embedded in paraffin, and 5 µm sagittal sections were probed for Ptgs2 (probe #316621) or Pim3 (probe #875481) using a RNAscope™ 2.5 HD Reagent Kit (Advanced Cell Diagnostics) and counterstained with hematoxylin.

### Analysis of published scRNA-seq datasets

We downloaded the raw sequencing data of mouse long bone cells and bone marrow stroma generated by Baryawno et al.[25] and Baccin et al.[24] from the SRA database under BioProjects PRJNA527721 and PRJNA505311. These data were preprocessed using CellRanger version 6.1.2 using mm10 reference genomes to generate count matrices. The count matrices were imported to Seurat software[67] version 4.1 for further analysis. Cells containing less than 10% of mitochondrial reads or less than 3500 genes were excluded from further analysis to eliminate potential dead cells or doublets. All data were harmonized to minimize the batch effect using Canonical Correlation Analysis[67]. Principal component analysis (PCA) was performed on highly variable genes of 6,000 cells to aggregate cells based on the similarities of their transcriptional profile. The results were used as input for clustering using Louvain algorithm with multilevel refinement and Uniform Manifold Approximation and Projection (UMAP) for dimension reduction. Only mesenchymal cells, identified by the expression of *Pdgfra*, were kept for further analysis. To identify upregulated specific gene markers of individual clusters we used the function FindMarkersAll using MAST algorithm[68]. Selected markers used to identify cell types were visualized using weighted kernel density estimation of expression level using Nebulosa package (version 1.8.0)[69]. The upregulated specific gene markers with a log$_2$ fold change greater than 0.5 and adjusted $p < 1^{e-50}$ were used for integrated analysis of metabolic pathways. To identify metabolic hot spots based on transcriptional regulation, called reporter metabolites[26], we integrated the gene specific markers on the mouse genome-scale metabolic model iMM1865[70] using piano R package (version 2.14.0)[71]. In brief, the genome-scale metabolic model was converted to a metabolic graph (unipartite), connecting metabolites and enzymes. Then, individual metabolite node in the metabolic graph are scored (enrichment p values) based

on transcripts encoded by their neighboring enzymes, following the method described by Patil and Nielsen[26]. The selected reporter metabolites of individual cell types were illustrated by dot plots. The relative expression of the metabolic marker genes (subset of the identified marker genes that are defined as metabolic genes) of individual cell types were summarized as a heatmap plot.

### Single cell RNA-sequencing and data analysis

Femur and tibia growth plates were dissected from 2 control and 3 Nampt[ΔPrx1] mice at P2. These mice were born from dams that received NR supplementation, as described above. Cells were dissociated enzymatically with 0.25% trypsin-EDTA for 30 min, followed by Liberase TM (Sigma/Roche; 1 Wunch unit/mL) for 2 h with continuous mixing, collecting released cell fractions every 30 min. The pooled cell fractions were then depleted of hematopoietic cells using a miniMACS platform using LS columns and a mouse Lineage Cell Depletion Kit (Miltenyi Biotec). Approximately 12,000 cells per genotype were encapsulated using a Chromium Controller (10X Genomics, Pleasanton, CA), and libraries were constructed using a Chromium Single Cell 3' Reagent Kit (10X Genomics). The libraries were then sequenced using an Illumina NovaSeq 6000 to generate fastq files. These files were preprocessed and analyzed as described in the "Analysis of published scRNA-seq" section above. The key results were plotted in dot-heat plot of directional enrichment score (-Log$_{10}$Pval). The single-cell data set generated in this publication is accessible in the NCBI's Sequence Read Archive (SRA) deposited under BioProject PRJNA914642.

### Statistical analysis

All numerical results are reported as mean ± standard deviation of the mean (SD). The data were analyzed by Student's t test (independent samples, two-sided) using GraphPad Prism 8 from GraphPad Software, after determining that the data were normally distributed and exhibited equivalent variances. Experiments using primary cell cultures were performed with at least three replicates and independently repeated two times.

### Reporting summary

Further information on research design is available in the Nature Portfolio Reporting Summary linked to this article.

## Data availability

The scRNA-seq data generated in this study have been deposited in the SRA database under accession code BioProject PRJNA914642. We also used scRNA-seq data generated by Baryawno et al.[25] and Baccin et al.[24] from the SRA database under BioProjects PRJNA527721 and PRJNA505311. Source data are provided with this paper.

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

## Acknowledgements
We thank J.A. Crawford, S.B. Berryhill, and H. Wu for technical support and Dr Shin-ichiro Imai for generously sharing the Nampt[fl] mice. This work was supported by US National Institutes of Health R01AR056679 (to MA), R01AG068449 (to MA and CAO), and P20GM125503 (to CAO) and the UAMS Bone and Joint Initiative.

## Author contributions
M.A. conceived and designed the experiments; A.W. and O.R.C. performed all animal experimentation; A.M.C., L.B.G., H.N.K., and M.M.A. generated and analyzed the in vitro data; O.R.C., R.M.P., and R.M. performed histological analysis; L.B.G and R.M.P. collected and processed samples for scRNA-seq; C.A.O. and I.N. analyzed scRNAseq data; R.M., E.S., and R.M.P. provided methodology and technical support; R.M.P. and M.A. wrote the manuscript; all authors reviewed the manuscript.

## Competing interests
The authors declare no competing interests.
