## [Peer Review File · Nature Communications]

REVIEWER COMMENTS

Reviewer #1 (Remarks to the Author):

Nampt mediated NAD biosynthesis is the predominant NAD⁺ biosynthesis pathway in mammals. Constitutive global deletion of NAMPT gene is embryonic lethal while inducible global and neuron-specific deletions in adult stage also lead to mouse death. Deletions of NAMPT in skeletal muscle and photoreceptors cause degeneration and deformities of respective tissues/cells. These studies demonstrated that NAMPT is an essential survival gene and deletion of NAMPT in specific tissues/cell types will cause significant outcomes. Using mesenchymal cell-specific NAMPT knockout mice, the current study adds additional evidence that NAMPT (and the NAD salvage pathway) plays a critical role in skeletal development and bone formation and growth during development. In addition, administration of NR can rescue skeletal defects in the NAMPT KO mice.

Major comments:

- Should provide immunostaining and west blot data to confirm the NAMPT reduction in KO mice.
- It is better to provide joint image in Fig 3 like in Fig 2.
- NAD is important factor for TCA cycle and glycolysis, it is necessary to do OCR and ECAR in chondrocytes from KO mice to study mitochondrial function/ respiration and glycolytic metabolism.
- Suggest doing some biochemical study based on the results of Fig 6 or NAD⁺ related enzymes such as Sirtuins for more in-depth mechanistic study.
- Fig 5B only included image of KO mice, an image of control mice should be included as well.
- Fig 5D is calcein-labeled image, it looks like other color as well, please label it.
- Tunel staining images in Fig 2B and Fig 4B are poor, high-resolution images should be provided.
- The metabolomic data in Fig 6B should be discussed in more detailed.
- The method part needs include the detail of metabolite analysis in Fig 6B
- The manuscript should cite the following 2 papers:
Wang et al. Cell reports (2017) Deletion of Nampt in Projection Neurons of Adult Mice Leads to Motor Dysfunction, Neurodegeneration, and Death 2): 2184-2200.
Zhang et al. Cell Death Dis. 2017:e2705. doi: 10.1038/cddis.2017.132.
- Most text in the scale bars in the figures are tool small, please make proper changes.

Minor

-Some figures have boxed region such as Fig 4A (control), but some does not have such as Fig 4A (KO mice), Fig 3D.

-Please provide full names for different cell types in Fig 6A-B, e.g., what is the name for growth plate chondrocytes in the figures?

-Fig 6A-B should be Fig 7A-B in line 189

-Some minor grammar issues, e.g., '10 cells types' in line 156 should be '10 cell types', '5 μ m (line 346)' should be 5 μ m (with space), 'maker genes' in line 422 should be marker genes

Reviewer #2 (Remarks to the Author):

In this manuscript, Aaron Warren et al. report that the NAD salvage pathway in mesenchymal cells is indispensable for skeletal development. They show that the inactivation of Nampt, a critical enzyme in the NAD salvage pathway, in all mesenchymal lineage cells of the limbs, result in dramatic limb deformities of new born mice. They observe that administration of the NAD precursor NR during pregnancy rescue the majority of in utero defects.

This is a potentially interesting study, which addresses an important aspect of the NAD salvage pathway in skeletal development. Some of my concerns are as below:

1. Provide the full names for NA, NAM, NMN and NR as well as abbreviations, where they are first mentioned in the manuscript.
2. In figure 4D, there is no statistical analysis for quantitative RT-PCR. In addition, in line 398, the author stated that target gene expression was calculated by normalizing to the housekeeping gene ribosomal protein S2, Mrps2. However, in figure 4D, Mrps7 was used. Please double check which one is correct.
3. In figure5C, specific osteoblast makers, such as osteocalcin, should be used to observe osteoblasts. In addition, the quantification of the number of osteoblasts and mineral apposition rate (MAR) should be supplied.
4. The author should describe Figure7D-F in more detail in the figure legend.
5. In the NR rescue experiments, mice were treated with 24mM NR (Tru Niagen, ChromaDex) administered ad libitum in the drinking water in light-protected bottles. Why they choose 24mM NR in the rescue experiments? Whether they find concentration dependent differences?

(Rebuttal letter)

We thank the reviewers for their comprehensive review and their thoughtful criticisms, and the editorial team for the opportunity to submit a revised manuscript. We have addressed each of the concerns of the reviewers, and believe that these revisions have improved the manuscript. The revised portions of the text are yellow-highlighted in the manuscript. Due to the extensive amount of work performed to address the comments, we added one more author (Landon Gatrell) and changed the original order of the authors to better reflect their contributions to the manuscript.

REVIEWER COMMENTS

Reviewer #1:

Nampt mediated NAD biosynthesis is the predominant NAD⁺ biosynthesis pathway in mammals. Constitutive global deletion of NAMPT gene is embryonic lethal while inducible global and neuron-specific deletions in adult stage also lead to mouse death. Deletions of NAMPT in skeletal muscle and photoreceptors cause degeneration and deformities of respective tissues/cells. These studies demonstrated that NAMPT is an essential survival gene and deletion of NAMPT in specific tissues/cell types will cause significant outcomes. Using mesenchymal cell-specific NAMPT knockout mice, the current study adds additional evidence that NAMPT (and the NAD salvage pathway) plays a critical role in skeletal development and bone formation and growth during development. In addition, administration of NR can rescue skeletal defects in the NAMPT KO mice.

Major comments:

-Should provide immunostaining and west blot data to confirm the NAMPT reduction in KO mice.

We now provide immunostaining (Fig. 3D) and Western blot data (Supplementary Fig. 7) to confirm Nampt deletion.

-It is better to provide joint image in Fig 3 like in Fig 2.

The picture of the joint in Fig. 3 has been modified to illustrate the levels of NAMPT, GAGs, and dead cells in the knee joints of P2 mice that were rescued with NR. The humerus pictures of the original Fig.3 are now enlarged and depicted in Supplementary Fig. 2.

-NAD is important factor for TCA cycle and glycolysis, it is necessary to do OCR and ECAR in chondrocytes from KO mice to study mitochondrial function/ respiration and glycolytic metabolism.

Because the major source of energy in growth plate chondrocytes is glycolysis, we performed ECAR using chondrocyte cultures from newborn Nampt cKO and littermate controls. As shown in the new Supplementary Fig. 7, we found that ECAR is decreased in cultured cells from KO mice, along with a decrease in ATP levels. Nonetheless, because the culture conditions do not replicate the poor nutrient supply experienced by the cells in the growth plate, it is very likely that the changes observed in the cultures are of a lower

magnitude than the ones occurring in vivo. This discussion is now included in the Results (Ln 317).

-Suggest doing some biochemical study based on the results of Fig 6 or NAD⁺ related enzymes such as Sirtuins for more in-depth mechanistic study.

To address the potential cause of chondrocyte death as a consequence of Nampt deletion, we performed scRNAseq in freshly isolated growth plates from P2 mice of dams that were fed NR during pregnancy. We found profound changes in gene expression which indicated suppression of multiple metabolic pathways and an overall decrease in translation. We propose that the range of metabolic disturbances occurring simultaneously likely contributes to chondrocyte death in Nampt^{ΔPrx1} mice. As the reviewer suggested, we also examined potential decreases in Sirtuin and Parp expression and activity but found no strong evidence to suggest that these enzymes are greatly impacted in chondrocytes from mice with Nampt deletion.

-Fig 5B only included image of KO mice, an image of control mice should be included as well. **An image of control mice has been added to Fig. 5B.**

-Fig 5D is calcein-labeled image, it looks like other color as well, please label it. **We replaced the initial images with new ones showing the same intensity of calcein labels. As suggested, the calcein labels are now indicated using arrows.**

- TUNEL staining images in Fig 2B and Fig 4B are poor, high-resolution images should be provided.

Higher magnifications of the TUNEL images are now provided in Figs 2B and 4B.

-The metabolomic data in Fig 6B should be discussed in more detailed.

A more detailed discussion of the reporter metabolite data is now provided in lines 175-177 and 185-190.

-The method part needs include the detail of metabolite analysis in Fig 6B

The Methods' section now includes a more detailed description of the reporter metabolite analysis (line XX). (NOT DONE YET)

-The manuscript should cite the following 2 papers:

Wang et al. Cell reports (2017) Deletion of Nampt in Projection Neurons of Adult Mice Leads to Motor Dysfunction, Neurodegeneration, and Death 2): 2184-2200.

Zhang et al. Cell Death Dis. 2017:e2705. doi: 10.1038/cddis.2017.132.

We apologize for not having included the papers above in the original submission, as these are very relevant to the work presented in the current manuscript. We now cite the papers in lines 357 and 415.

-Most text in the scale bars in the figures are too small, please make proper changes.

We have enlarged the text in the scale bars, as requested.

Minor

-Some figures have boxed region such as Fig 4A (control), but some does not have such as Fig 4A (KO mice), Fig 3D.

Fig 3D now includes a higher magnification boxed region.

-Please provide full names for different cell types in Fig 6A-B, e.g., what is the name for growth plate chondrocytes in the figures?

The only name we abbreviated was Osteoblast_prog. We now provide the full name (Osteoblast_progenitors) for this cell type in Fig 6A. All other names for the different cell types are the ones defined in the cited publications (Baccin et al, 2020; Baryawno et al, 2019) and are not abbreviations. We respectfully submit that we should maintain the original names not to create any confusion. The cells named chondrocytes are the growth plate chondrocytes.

-Fig 6A-B should be Fig 7A-B in line 189

This mistake has been corrected.

-Some minor grammar issues, e.g., '10 cells types' in line 156 should be '10 cell types', '5µm (line 346)' should be 5 µm (with space), 'maker genes' in line 422 should be marker genes.

All these mistakes have been corrected.

Reviewer #2 (Remarks to the Author):

In this manuscript, Aaron Warren et al. report that the NAD salvage pathway in mesenchymal cells is indispensable for skeletal development. They show that the inactivation of Nampt, a critical enzyme in the NAD salvage pathway, in all mesenchymal lineage cells of the limbs, result in dramatic limb deformities of new born mice. They observe that administration of the NAD precursor NR during pregnancy rescue the majority of in utero defects.

This is a potentially interesting study, which addresses an important aspect of the NAD salvage pathway in skeletal development. Some of my concerns are as below:

1. Provide the full names for NA, NAM, NMN and NR as well as abbreviations, where they are first mentioned in the manuscript.

All full names followed by abbreviations for NA, NAM and NR had been provided in the Introduction of the original submission. We now include the full name for NMN in line 109.

2. In figure 4D, there is no statistical analysis for quantitative RT-PCR. In addition, in line 398, the author stated that target gene expression was calculated by normalizing to the housekeeping gene ribosomal protein S2, Mrps2. However, in figure 4D, Mrps7 was used. Please double check which one is correct.

We apologize for these mistakes, the p values relative to the comparisons performed in figure 4D are now provided, as is the correct name (Mrps2) for the housekeeping gene.

3. In figure5C, specific osteoblast makers, such as osteocalcin, should be used to observe osteoblasts. In addition, the quantification of the number of osteoblasts and mineral apposition rate (MAR) should be supplied.

As suggested by the reviewer, we now provide the number of osteoblasts and MAR in figure 5. However, due to the major differences in size and shape of the bones it is difficult to accurately evaluate possible consequences of Nampt deletion in osteoblastic cells. To examine the contribution of Nampt in osteoblasts during bone development, we deleted Nampt in Osx1-Cre targeted cells which include all osteoblast precursors, osteoblasts, and osteocytes as well as a small number of proliferating and hypertrophic chondrocytes. We show in Fig. 5E-G that Nampt^{ΔOsx1} mice were grossly undistinguishable from Osx1-Cre control littermates up to 4 weeks of age. At this age, body weight and femur and total BMD were similar between the two genotypes, in stark contrast with the findings obtained with Nampt^{ΔPrx1} mice (Fig. 1). Together, these results clearly demonstrate the different dependency on Nampt of chondrocytes versus osteoblasts during bone development.

4. The author should describe Figure7D-F in more detail in the figure legend.

We added more detail to the figure legend to better explain the culture conditions of panels D-E.

5. In the NR rescue experiments, mice were treated with 24 mM NR (Tru Niagen, ChromaDex) administered ad libitum in the drinking water in light-protected bottles. Why they choose 24mM NR in the rescue experiments? Whether they find concentration dependent differences?

The dose of NR used was based on published studies by multiple labs. In these studies, the doses administered to mice range from about 200-1000 mg/kg administered by a variety of means (water, diet, gavage, osmotic pumps, and IP or SC injections). We initially tried a dose equivalent to 400 mg/kg but this dose was not sufficient to rescue the bone phenotype. We then increased the dose to 650 mg/kg which greatly attenuated the bone defects. The information about the lack of effect of a lower dose of NR is now added to the manuscript in the Result section (line 110).

REVIEWER COMMENTS

Reviewer #1 (Remarks to the Author):

The authors have made a great deal of effort to address the critics from both reviewers and improve the manuscript, but there are a few concerns need to be addressed/clarified to further improve manuscript.

-From text, in Fig 3, $Nampt\Delta Prx1$ mice were administered with NR, it is better to label as like $Nampt\Delta Prx1+NR$ mice

-It would better to do more quantitative analysis of TUNEL staining to compare $Nampt\Delta Prx1$ mice (Fig 2B) and $Nampt\Delta Prx1+NR$ (Fig 3F).

Notice $Nampt\Delta Prx1$ mice died a few days after birth, it is not possible to compare in adult stage $Nampt\Delta Prx1$ mice with $Nampt\Delta Prx1+NR$, but the effect of NR should be compared between $Nampt\Delta Prx1$ mice and $Nampt\Delta Prx1+NR$ mice at the same age, not between control ($NAMPTf/+$) mice & $Nampt\Delta Prx1+NR$ mice (like in Fig 3 and 4), the rationale needs to be clarified.

Is it possible to have a survival curve between $Nampt\Delta Prx1$ mice and $Nampt\Delta Prx1+NR$, this will be very helpful to understand the therapeutic effect of NR?

-While you have immunostaining of $Nampt\Delta Prx1+NR$ mice in Fig 3D, it is better to have immunostaining of $Nampt\Delta Prx1$ mice without NR. This data can be put in Fig 1 as a part of characterization of $NAMPT$ KO mice in the beginning.

Are other enzymes $NMNAT1-3$ in salvage pathway affected in $Nampt\Delta Prx1$ mice?

It is desirable to discuss possible pathological features if $NAMPT$ is deleted in adult mice using inducible $Prx1-CreERT2$ driver line?

Minors

Please put scale bar in Fig 3E-F

Western blot data actually is in Supp Fig 9, not Suppl Fig 7.

ECAR is in Suppl Fig 9, not Suppl Fig 7

Reviewer #2 (Remarks to the Author):

It is OK.

We thank the reviewers for their positive comments on our revised manuscript. We have now addressed the remaining comments from Reviewer 1 and believe that these revisions have improved the manuscript. The revised portions of the text are yellow-highlighted in the manuscript.

Reviewer #1 (Remarks to the Author):

The authors have made a great deal of effort to address the critics from both reviewers and improve the manuscript, but there are a few concerns need to be addressed/clarified to further improve manuscript.

-From text, in Fig 3, $Nampt\Delta Prx1$ mice were administered with NR, it is better to label as like $Nampt\Delta Prx1+NR$ mice.

This is a good suggestion. We took it a step further and used “ $Nampt\Delta Prx1+pNR$ ” in Figures 3, 4, 5, 8, and 9 to remind the readers that the NR rescue is prenatal only.

-It would better to do more quantitative analysis of TUNEL staining to compare $Nampt\Delta Prx1$ mice (Fig 2B) and $Nampt\Delta Prx1+NR$ (Fig 3F).

We have not performed a quantitative assessment of the TUNEL staining due to the following reasons: 1) in the experiment of figure 2B the morphology of the tissue and the cellular content is so different between the two genotypes that it would be very difficult to normalize the counting area for a proper comparison. 2) The number of TUNEL positive cells in the growth plates of control mice is negligible (that is, outside the hypertrophic zones). Performing statistical analysis in the situation where one group that has effectively zero observations is not the best practice. Instead, in order to provide the reader evidence for the reproducibility of cell death within the rescue experiment in Figure 3, we have provided additional images using knees of separate mice as well as representative images from the hip and ankle joints within new supplementary figure 4.

Notice $Nampt\Delta Prx1$ mice died a few days after birth, it is not possible to compare in adult stage $Nampt\Delta Prx1$ mice with $Nampt\Delta Prx1+NR$, but the effect of NR should be compared between $Nampt\Delta Prx1$ mice and $Nampt\Delta Prx1+NR$ mice at the same age, not between control ($NAMPTf/+$) mice & $Nampt\Delta Prx1+NR$ mice (like in Fig 3 and 4), the rationale needs to be clarified.

We apologize for the misunderstanding, but our intention was never to examine $Nampt\Delta Prx1$ at the adult stage. The comparison of the effects of $Nampt\Delta Prx1$ mice with $Nampt\Delta Prx1+NR$ at the same age were done at P2 ($Nampt\Delta Prx1$ in Fig 1-2 and $Nampt\Delta Prx1+NR$ in Fig. 3). The rationale for the work in Fig 4 and 5 was to examine the consequences of $Nampt$ deletion in the post-natal development of the growth plate. To this end, we took advantage of the fact that NR administration to dams during gestation rescued most of the $Nampt$ deletion phenotypes that occur in utero. The removal of NR post-birth allowed us to examine the dependency of the growth plate on $Nampt$ during epiphyseal bone development (P7 - P28). We respectfully submit that the use of $Nampt^{f/+}$ littermate mice as controls in these experiments is appropriate. We now provide a clearer rationale in the Result section (line 137) to explain our experiments.

Is it possible to have a survival curve between *Nampt* Δ *Prx1* mice and *Nampt* Δ *Prx1*+NR, this will be very helpful to understand the therapeutic effect of NR?

Our studies were not designed to perform survival curves. The cause of mortality of the *Nampt* Δ *Prx1* mice is the defective ambulation and consequent inability to nurse. This cause of mortality is not translatable into the human situation and we respectfully submit that it should not be used as an endpoint to understand the therapeutic effect of NR. Please, be reminded that these mice are born without forelimbs and extremely defective hindlimbs. All these mice die up to P2. In contrast, the mice that received pre-natal NR are almost normal at birth. Nonetheless, the limbs in these mice do not develop much further after birth because the rescue with NR was limited to the pre-natal period. The in utero rescue is sufficient to allow the mice to nurse and feed on their own at least until P28, the longest these mice were kept alive.

-While you have immunostaining of *Nampt* Δ *Prx1*+NR mice in Fig 3D, it is better to have immunostaining of *Nampt* Δ *Prx1* mice without NR. This data can be put in Fig 1 as a part of characterization of NAMPT KO mice in the beginning. We now provide immunostaining of *Nampt* in Fig. 1D.

Are other enzymes NMNAT1-3 in salvage pathway affected in *Nampt* Δ *Prx1* mice?

The expression of these enzymes was not impacted in *Nampt* Δ *Prx1* mice. This information is included in Supplementary Table 1.

It is desirable to discuss possible pathological features if NAMPT is deleted in adult mice using inducible *Prx1*-CreERT2 driver line?

This is a great suggestion and we are indeed pursuing studies to examine the role of *Nampt* in adult articular cartilage and the development of osteoarthritis. We inserted a short section in the Discussion addressing these considerations (line 376).

Minors

Please put scale bar in Fig 3E-F.

Scale bars have been added to all panels.

Western blot data actually is in Supp Fig 9, not Suppl Fig 7.

This mistake has been fixed.

ECAR is in Suppl Fig 9, not Suppl Fig 7

This mistake has been fixed.

Reviewer #2 (Remarks to the Author):

It is OK.

REVIEWERS' COMMENTS

Reviewer #1 (Remarks to the Author):

Authors addressed and clarified all the critics/comments. The only minor thing I identified is that there is no scale bar in the right panels of the Fig 4.